# IGF2BP3 promotes mRNA degradation through internal m⁷G modification

Chang Liu[1,2,5,8], Xiaoyang Dou[1,2,8], Yutao Zhao[1,2,8], Linda Zhang[1,2], Lisheng Zhang [1,2,6,7], Qing Dai [1,2], Jun Liu [3,4], Tong Wu [1,2], Yu Xiao[1,2] & Chuan He [1,2] ✉

Recent studies have suggested that mRNA internal m⁷G and its writer protein METTL1 are closely related to cell metabolism and cancer regulation. Here, we identify that IGF2BP family proteins IGF2BP1-3 can preferentially bind internal mRNA m⁷G. Such interactions, especially IGF2BP3 with m⁷G, could promote the degradation of m⁷G target transcripts in cancer cells. IGF2BP3 is more responsive to changes of m⁷G modification, while IGF2BP1 prefers m⁶A to stabilize the bound transcripts. We also demonstrate that p53 transcript, *TP53*, is m⁷G-modified at its 3'UTR in cancer cells. In glioblastoma, the methylation level and the half lifetime of the modified transcript could be modulated by tuning IGF2BP3, or by site-specific targeting of m⁷G through a dCas13b-guided system, resulting in modulation of cancer progression and chemosensitivity.

To date, over 150 modifications have been identified on cellular RNA, many of which are known to exist in eukaryotes[1]. Among them, *N⁶*-methyladenosine (m⁶A), discovered in eukaryotic mRNA in 1974, is the most prevalent internal modification in mRNA and long non-coding RNA (lncRNA)[2,3]. The modulators of RNA modifications can be cataloged into three groups: writers, erasers, and readers, which install, remove, and recognize the modifications. This new layer of regulation impacts many regulatory processes. For example, the m⁶A writer METTL3 has been related to both transcriptional and post-transcription regulations[4–7]. The depletion of erasers ALKBH5 or FTO notably impacts both nuclear and cytosolic regulations of mRNA processing and metabolism[8–10]. Various m⁶A-binding proteins bind the modification and modulate cellular localization, stability, and translation of the target transcripts. The interaction of m⁶A with its reader YTHDF2 destabilizes modified mRNAs while its interaction with YTHDF1 could affect translation[11,12].

The dysregulation of RNA modifications or their regulators has been associated with human diseases including cancer. For example,

demethylase FTO has been reported to be upregulated in *MLL*-rearranged AML, which promotes cell viability and leukemogenesis through the demethylation of functionally important genes[13]. In glioblastoma, the overexpression of m⁶A methyltransferase METTL3 or METTL14, or correspondingly, the knockdown of *FTO* inhibited the growth and self-renewal of glioblastoma stem cells[14]. The m⁶A reader IGF2BP2 was reported to interact with the methylation on lncRNA *DANCR* to stabilize the transcript, thus promoting cell growth in pancreatic cancer[15].

Apart from the well-known m⁶A, other mRNA modifications have also been identified and mapped within the transcriptome, such as internal *N⁷*-methylguanosine (m⁷G). m⁷G, an ubiquitous mRNA cap modification critical to mRNA life cycle[16–18], also exists as an internal modification within transfer RNA (tRNA)[19] and ribosomal RNA (rRNA)[20]. However, it was not until recently that internal m⁷G on mRNA has been identified and mapped in cancer cells[21–24]. The methyltransferase complex of tRNA m⁷G modification (METTL1-WDR4) was found to be responsible for a subset of mRNA internal m⁷G

[1]Department of Chemistry, Department of Biochemistry and Molecular Biology, and Institute for Biophysical Dynamics, The University of Chicago, Chicago, IL 60637, USA. [2]Howard Hughes Medical Institute, The University of Chicago, Chicago, IL 60637, USA. [3]State Key Laboratory of Protein and Plant Gene Research, School of Life Sciences, Peking-Tsinghua Center for Life Sciences, Peking University, Beijing 100871, China. [4]Beijing Advanced Center of RNA Biology (BEACON), Peking University, Beijing 100871, China. [5]Present address: Department of Genetics, School of Medicine, Stanford University, Stanford, CA 94305, USA. [6]Present address: Division of Life Science, The Hong Kong University of Science and Technology, Hong Kong, China. [7]Present address: Department of Chemistry, The Hong Kong University of Science and Technology, Hong Kong, China. [8]These authors contributed equally: Chang Liu, Xiaoyang Dou, Yutao Zhao. ✉e-mail: chuanhe@uchicago.edu

installation[21]. METTL1 has long been highlighted among the methyltransferase-like protein family as its high expression is notably associated with the high-grade tumors and poor prognosis[25,26]. Specifically, the high expression of METTL1, as the tRNA methyltransferase, could impact translation in glioblastoma and neuroblastoma through modulating levels of m7G-modified tRNAs[27,28]. A recent study identified QKI family proteins as the readers of internal m7G modification[29]. QKI proteins, especially QKI7, could shuttle m7G transcripts into stress granules under stress conditions and further impact decay and translation of these transcripts. These studies have suggested that internal m7G, together with its methyltransferase METTL1 and potential readers, is closely related to mRNA metabolism and cellular regulation processes such as cancer development.

Here, we present the IGF2BP proteins as the protein family that could preferentially bind internal m7G over G on mRNA in cancer cells. While IGF2BPs have been identified as m6A readers, the individual members display distinct preferences toward mRNA m6A and m7G modifications. The interaction of m7G with IGF2BP1 and IGF2BP3, especially the latter one with higher affinity, could promote the degradation of m7G-modified transcripts, whereas IGF2BP1 prefers m6A and stabilizes the bound transcripts. We then took advantage of the inactive dCas13b tethering system to specifically target individual m7G sites. The tethering of either IGF2BP3 or the methyltransferase catalytic core METTL1 would decrease the half lifetime of the target transcripts. We found that IGF2BP3 could promote the degradation of m7G target transcripts through its interaction with EXOSC2 of the exosome complex. Inspired by the high correlation of METTL1 with p53 related pathways in glioma, we mined and found that the p53 transcript, *TP53*, is m7G-modified at its 3'UTR by METTL1. We further tested in selected glioblastoma cells and demonstrated that the application of dCas13b tethering systems (IGF2BP3 or METTL1) on *TP53* could promote its decay and downregulate both the transcript and protein levels, which notably impact glioblastoma progression and chemotherapy resistance.

## Results

### METTL1 is involved in glioma and p53-related pathways and *TP53* methylation

METTL1 is highly expressed in various tumors compared to normal tissues (Supplementary Fig. 1a). It has been reported to play important roles in tumorigenesis or stemness maintenance[27,30–32]. A recent study showed that the upregulation of METTL1, as tRNA methyltransferase, could stabilize the m7G-modified tRNA and promote translation of oncogenic transcripts in neuroblastoma[28]. Immunohistochemical staining (IHC) data in other studies also demonstrated high protein expression levels of METTL1 in primary glioblastoma tissue compared to normal cerebral tissue[27]. Through mining TCGA datasets, we also noticed that, in glioma, the higher expression level of METTL1 is closely correlated to poorer survival rate (Fig. 1a). However, such effect of METTL1 expression on the survival rate was dramatically diminished in patients with *TP53* mutations (Fig. 1b and Supplementary Fig. 1b). We observed that METTL1 is highly involved in p53-related pathways including cell cycle and p53 signaling (Fig. 1c). In addition, the tumor survival dependence of WDR4, the METTL1 methyltransferase partner, is not only significantly correlated with that of METTL1 (Fig. 1d), but also shows close relationship with these pathways (Supplementary Fig. 1c).

*TP53*, one of the most frequently mutated gene in human cancer, is well known as the guardian of the genome[33]. We were curious how METTL1 expression correlates with glioma survival in a *TP53*-mutation dependent way, and whether METTL1 and WDR4, also as the writer for a subset of internal mRNA m7G modification, could affect tumor pathways through mRNA regulation. We therefore examined the methylation levels of the transcripts related to the p53 pathways based on the published m7G-MeRIP-seq datasets in HepG2 cells. Interestingly,

we found that the transcript of p53, *TP53*, is m7G-methylated in its 3' UTR, and the methylation peak overlaps well with the binding sites of METTL1 and WDR4 (Fig. 1e). The methylation level also decreased upon *METTL1* stable knockdown (Fig. 1f), which is reproducible upon the transient knockdown of *METTL1* (Fig. 1g), suggesting that METTL1 is involved in the m7G methylation at the 3' UTR of *TP53*.

As p53 is frequently mutated in glioblastoma[34] and the effect of METTL1 expression on the survival rate of glioma/glioblastoma could be dramatically affected by p53 mutation (Fig. 1b and Supplementary Fig. 1b), we selected several representative glioblastoma cell lines with different *TP53* status to study the effects of m7G methylation on *TP53* transcript[35–37]: U87MG and A172 express wildtype p53. LN229 has a P98L mutation in the proline-rich domain[36], which was reported to remain a partial wild-type function[38]. T98G cells express a p53 mutant protein with M237I mutation that severely compromises the transactivation capacity of the wild-type protein[39–41]. It has also been reported that cells expressing the M237I-p53 mutant have a lower chemosensitivity to temozolomide (TMZ) treatment compared to the wild-type[42]. We first quantified the relative m6A or m7G modification levels of the entire *TP53* transcript using MeRIP enrichment followed by qPCR quantification and observed little difference (Supplementary Fig. 1d). However, when we examined the methylation level at the 3' UTR, we noticed huge variations across different cell lines. While no m6A enrichment was observed for all cell lines, LN229 and U87MG presented the highest m7G modification levels, and the other two cell lines had low m7G when compared to HepG2 cells we used as the reference (Fig. 1h). We also noticed that the mRNA level of *TP53* varied quite notably across different cell lines (Fig. 1i), so as its protein expression levels (Supplementary Fig. 1e), and surprisingly, the m7G methylation level at the 3' UTR displayed a negative correlation with the *TP53* transcript abundance (Fig. 1j). We didn't observe much correlation of the protein levels of m7G regulators (METTL1, WDR4) with the *TP53* transcript level, as these proteins remain similar in the expression levels across tested cell lines (Supplementary Fig. 1e). We proceeded to ask whether there are potential m7G binding proteins that may explain the observed negative correlation between the internal m7G level on the target mRNA and the transcript abundance.

### IGF2BP family proteins identified as the readers of internal m7G modification

We started with pulling down potential binding proteins of mRNA internal m7G in HepG2 cells. As m7G is not compatible with oligo synthesizer, we designed a probe with only one G in the DNA template and prepared RNA probes using either unmodified GTP or m7GTP during in vitro transcription. The probes were then biotinylated at the 3' end and incubated with cell lysate to pull down interactive proteins, followed by protein mass spectrometry identification (data available via ProteomeXchange with identifier PXD049390, proteins enriched by m7G probes listed in Supplementary Data 1). Our results highlighted IGF2BP family proteins as the ones showing preference to internal m7G over G in RNA (Fig. 2a). The recently reported reader QKI[29] was also observed.

IGF2BPs are important RNA binding proteins actively involved in RNA metabolism and diseases[43–45]. Previous studies have identified IGF2BPs as m6A readers to stabilize m6A transcripts and regulate translation[46]. To validate our mass spectrometry results, we performed western blot to verify the binding affinity of IGF2BPs on m7G (vs. G) probes (Fig. 2b) and m6A (vs. A) probes in parallel (Supplementary Fig. 2a). Though similar in protein structural arrangements, the three members differed slightly in the binding affinity of different modifications. IGF2BP1 appears to prefer m6A, while m7G was more favored by IGF2BP2 and IGF2BP3. The m7G probes were also more favorably bound by these proteins compared to the G-bearing probes in the electrophoresis mobility shift assay (EMSA) (Supplementary Fig. 2b). We further quantified the internal m7G/G levels of RNA bound by IGF2BP proteins through in vitro pulldown in HepG2 cell lysate using

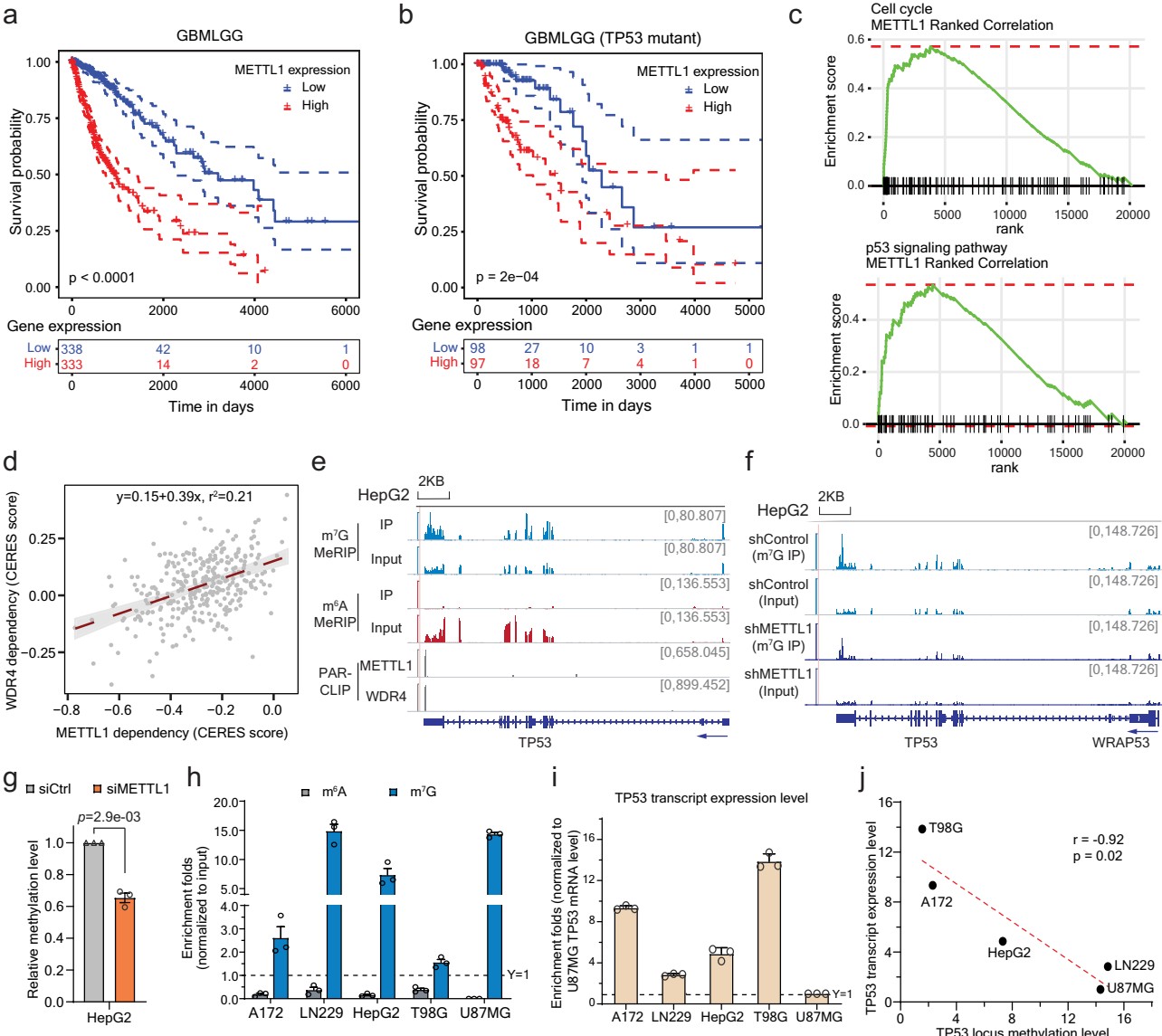

**Fig. 1 | METTL1 may regulate *TPS3* abundance through its 3'UTR m⁷G methylation in cancer cells.** **a**, **b** Kaplan-Meier survival analysis in TCGA database for Glioma (TCGA-GBMLGG): the entire dataset (**a**), and cases with mutant *TPS3* (**b**). The patients were divided into two groups of equal size based on the *METTL1* levels. The ones lower than the median value are all grouped into the "Low" group while the ones higher than the median value are all grouped into "High". *P*-value detected by log-rank test. **c** Gene Set Enrichment Analysis (GSEA) of genes in 'cell cycle' and 'p53 signaling pathway' KEGG pathways against ranked list of genes according to *METTL1* expression level. **d** The correlation of CERES dependency score between *METTL1* and *WDR4*, with standard error marked in gray. The CERES was developed to estimate gene-dependency level from CRISPR-Cas9 essentiality screens while accounting for the effects[80]. **e** The integrative genomics viewer (IGV) plots showing the m⁷G-MeRIP-seq peaks at the 3' UTR end of *TP53*, which also overlap with METTL1 and WDR4 PAR-CLIP peaks. A minor m⁶A peak was detected in the distant CDS region adjacent to 5'UTR region. Y-axis showing counts per ten million reads.

**f** IGV plots showing the m⁷G-MeRIP-seq peaks at the 3' UTR end of *TP53* upon stable knockdown of *METTL1* compared to control in HepG2 (GSE112276). Y-axis showing counts per ten million reads. **g** Relative m⁷G methylation levels of the *TP53* 3' UTR locus upon *METTL1* knockdown in HepG2 cells, normalized to the methylation level in the knockdown control cells. Mean ± SEM (*n* = 3) with two-tailed Student's *t*-tests. **h** Relative m⁶A (gray) and m⁷G (blue) methylation levels of the 3' UTR region of *TP53* based on MeRIP enrichment followed by qPCR quantification. Mean ± SEM of three independent experiments. **i** The mRNA levels of *TP53* in different cell lines. For each cell, the *TP53* expression level was normalized to total RNA amount respectively. And the comparison across the cell lines were normalized to U87MG. Mean ± SEM of three independent experiments. **j** Spearman correlation of the 3' UTR m⁷G methylation level and the mRNA abundance of *TP53*. Significance was determined by two-sided Pearson's correlation test. All source data are provided as a Source Data file.

corresponding antibodies and compared those to that of the input. The m⁷G modification levels were roughly doubled in all three IGF2BP protein-bound fractions (Fig. 2c). Such results remained similar when we overexpressed METTL1 in HepG2 cells; overexpressed cells showed higher internal m⁷G/G levels upon all three IGF2BP pulldown, suggesting that IGF2BP family members could all recognize METTL1 target transcripts (Supplementary Fig. 2c). Altogether, our results suggest preferential binding of RNA internal m⁷G by IGF2BP proteins.

The structures of IGF2BPs have been well studied. They share a characteristic arrangement of six canonical RNA binding domains, namely two consecutive RNA recognition motifs (RRMs) followed by four hnRNP K homology (KH) domains, where RRM1-2, KH1-2, and KH3-4 are arranged into pairs[47,48]. Previous studies have shown that the KH3-4 domains are critical for IGF2BPs binding to m⁶A: mutated KH3-4 domains abolished the binding of all three proteins with m⁶A[46]. To evaluate IGF2BPs' binding of internal m⁷G, we overexpressed FLAG-

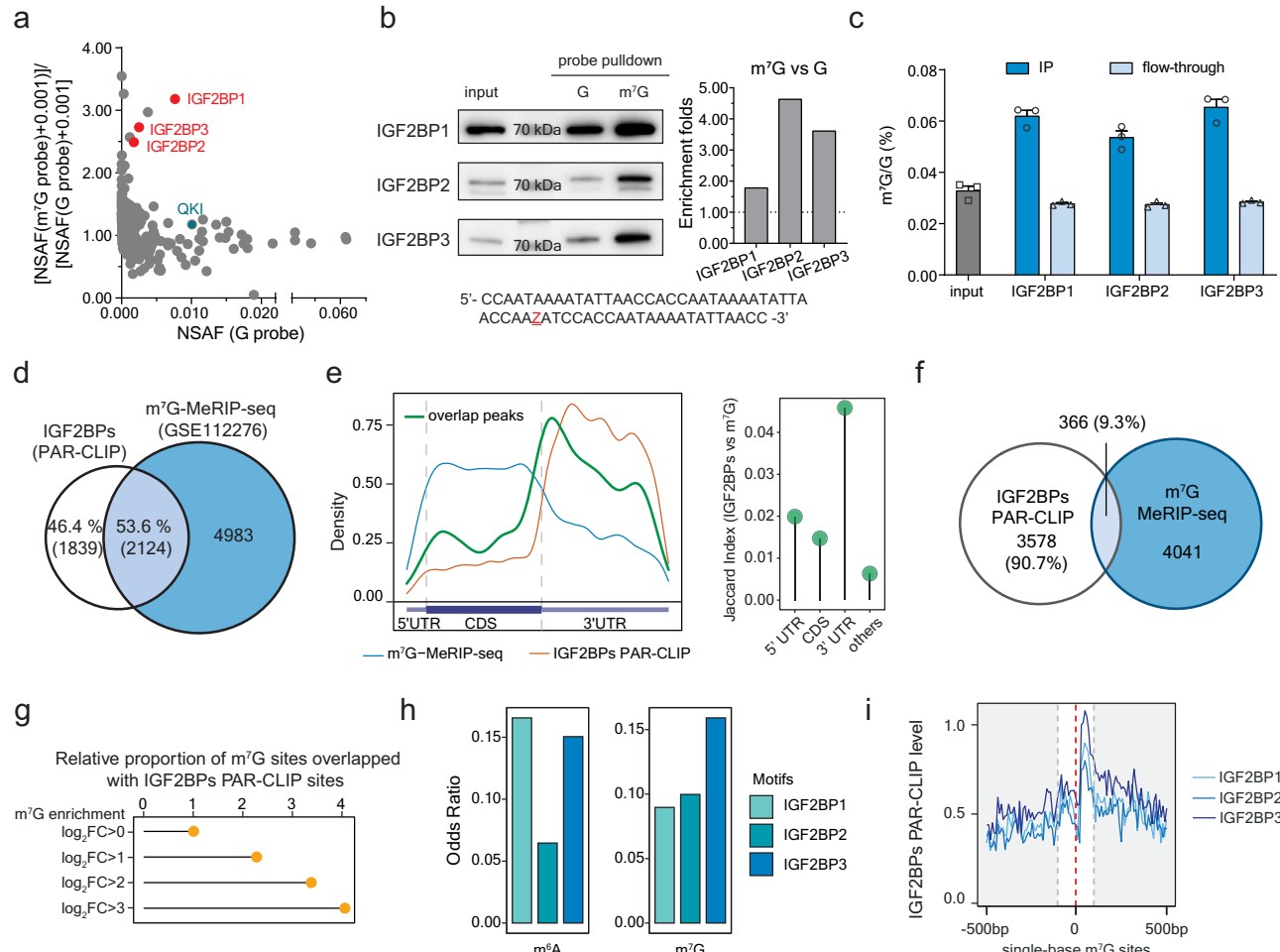

**Fig. 2 | IGF2BP family proteins bind to mRNA internal m⁷G. a** Protein mass spectrometry results with x-axis showing general binding on unmodified RNA and y-axis showing the binding of m⁷G over G. To calculate enrichment ratios for proteins identified by only one probe, 0.001 was added to all normalized spectral abundance factor (NSAF). The IGF2BP family proteins are highlighted in red. The reported reader QKI is marked in green. Top-ranked proteins are listed in Supplementary Data 1. **b** Western blot of HepG2 cell lysate in vitro pulldown using the m⁷G probe and G probe, respectively, with the estimation of enrichment folds plotted on the right. This experiment was repeated independently twice with similar results. The probe sequence is listed below the blots. **c** LC-MS/MS quantification of m⁷G/G levels showing enrichment of m⁷G after immunoprecipitation using antibodies against IGF2BPs. Mean ± SEM of three independent experiments.

**d** Venn diagram of overlaps of transcripts that were bound by IGF2BPs and modified with m⁷G. **e** Left: metagene plot of IGF2BPs binding sites (not overlapped with m⁷G modified sites) (orange), m⁷G modified sites (not overlapped with IGF2BPs binding sites) (blue) and their overlapping sites (green). Right: Lollipop plot showing the overlaps between m⁷G and IGF2BPs quantified using Jaccard Index. **f** Venn diagram of overlaps of peaks that were bound by IGF2BPs and modified with m⁷G. **g** Proportion of m⁷G sites overlapping with IGF2BPs binding sites. m⁷G sites were categorized into groups based on m⁷G enrichment quantified as log₂FC(IP/Input). **h** Enrichment of m⁶A (left) and m⁷G (right) with IGF2BP1, IGF2BP2, and IGF2BP3 motifs, respectively, quantified with odds ratio. **i** Binding intensity of IGF2BP1, IGF2BP2, and IGF2BP3 at m⁷G sites and their flanking 500 bp, respectively. All source data are provided as a Source Data file.

tagged IGF2BP mutants with truncated RRM domains or mutated KH domains and tested their binding towards m⁷G probes (Supplementary Fig. 2d). The KH mutations refer to the full-length protein with mutations of GxxG to GEEG in the KH domains as previously described[46]. The cell lysate was incubated with m⁷G-modified biotinylated probes, and the binding intensity was examined by pulldown and western blot, and further quantified with comparison to the enrichment levels in the wild-type cell lysate. The mutation on KH3-4 domains almost abolished the binding affinity, suggesting that KH3-4 domains are critical to the modification binding, while the RRM domain did not contribute much. A slight difference occurred in the case of IGF2BP3. When KH3-4 domains of IGF2BP3 were mutated, this protein still displayed a slight binding towards the m⁷G modification, implying that the KH1-2 domains might also assist the m⁷G binding in IGF2BP3. In correspondence to the variations in binding affinities toward m⁶A over A or m⁷G over G (Fig. 2b and Supplementary Fig. 2a), our studies of different

binding domains suggested that IGF2BPs might bind m⁶A and m⁷G in a similar but not identical way. The structural and most likely electrostatic differences of these two modified bases might explain these variations, which await further structural investigations of the bound protein complexes.

### Transcriptomic binding of IGF2BP family proteins on internal m⁷G modification

To better understand the binding between m⁷G and IGF2BPs and potential impacts at the transcriptome level, we performed photoactivatable ribonucleoside-enhanced crosslinking and immunoprecipitation (PAR-CLIP) of the three IGF2BPs in HepG2 cells and overlapped their binding sites with m⁷G-MeRIP-seq, and in parallel with m⁶A-MeRIP-seq as a reference. Consistent to the previous report[46], over two-thirds of IGF2BPs-bound mRNAs were m⁶A-modified (Supplementary Fig. 3a). More than half of the IGF2BPs targets are modified

by internal m⁷G (Fig. 2d), comparably among all three members (Supplementary Fig. 3b).

IGF2BPs are regulators mainly bound at the 3′ UTR of its target transcripts (Fig. 2e). The metagene profiles confirmed overlap of the m⁷G peaks with IGF2BP-bound sites (Fig. 2e left in green), with most overlapping sites at the 3′ UTR compared to the non-overlapped but methylated ones (Fig. 2e left in blue), similar to the case of m⁶A modification (Supplementary Fig. 3c). Such patterns are conserved across three IGF2BP members (Supplementary Fig. 3d and 3e). When examining their binding overlaps at the 3′ UTR we observed about 10% of the IGF2BP binding sites adjacent to m⁷G peaks (Fig. 2f and Supplementary Fig. 3f), close to that around the m⁶A peaks (Supplementary Fig. 3g). Correspondingly, m⁷G sites with higher methylation levels showed better overlap with IGF2BP binding sites (Fig. 2g and Supplementary Fig. 3h).

Structural studies have shown that the KH1-2 and KH3-4 domains of IGF2BPs are prearranged to recognize bipartite RNA sequence motifs, with certain variations among the three members[47–49]. For example, IGF2BP3 has been found to prefer motifs such as GGCA-N$_{(9-25)}$-CA or CGGC-N$_{(15-25)}$-CA[49]. We therefore calculated the possibility that an m⁶A or m⁷G peak could cover the consensus favorable motifs of each IGF2BP protein. m⁷G transcripts are particularly favored by IGF2BP3 while m⁶A are more preferred by IGF2BP1 and IGF2BP3 (Fig. 2h), further supporting the difference of IGF2BP3 binding on m⁷G versus m⁶A.

In addition to m⁷G-MeRIP-seq, we took advantage of m⁷G-seq at base resolution to study the binding of IGF2BPs. The plots with more quantitative m⁷G-seq dataset also precisely demonstrated that IGF2BPs, especially IGF2BP3, bind at m⁷G sites (Fig. 2i). As a reference, WDR4, the RNA binding partner in the internal m⁷G writer complex, exhibited a similar pattern (Supplementary Fig. 3i). Altogether, we found that IGF2BPs interacts with internal m⁷G located at 3′ UTR of mRNA in HepG2 cells.

## IGF2BP1 and IGF2BP3 promote decay of the m⁷G-modified transcripts

IGF2BPs have been shown to stabilize m⁶A transcripts and play important roles in cancer development and prognosis[45,46,50–52]. We asked if IGF2BPs could affect the stability of m⁷G-modified transcripts. We performed transient knockdown of the three IGF2BP proteins (Supplementary Fig. 4a) followed by the RNA lifetime sequencing, with data highly consistent between each pair of replicates (Supplementary Fig. 4b). m⁶A targets were verified to be destabilized (Supplementary Fig. 4c). However, when it comes to m⁷G transcripts, the half lifetime increased upon the knockdown of *IGF2BP1* and *IGF2BP3* (Fig. 3a). The stabilization effect was positively correlated to the enrichment of both m⁷G modification levels (Fig. 3b) and IGF2BPs binding intensities (Supplementary Fig. 4d). We further verified the methylation levels (Fig. 3c) within the full-length transcripts of representative genes (Supplementary Fig. 4e) and confirmed the increased half-lifetime upon *IGF2BP1* and *IGF2BP3* knockdown, respectively (Fig. 3d and Supplementary Fig. 4f).

METTL1-WDR4, the tRNA m⁷G methyltransferase complex, has also been shown to act as mRNA writer for a subset of internal m⁷G modification in cancer cells. To further validate the effects of IGF2BP1 and IGF2BP3 on m⁷G-modified transcripts, we grouped the METTL1 targets according to their methylation changes upon *METTL1* knockdown. The transcripts with more decreases in their m⁷G methylation levels were more responsive to *IGF2BP1* and *IGF2BP3* perturbation, and thus were more stabilized upon knockdown (Supplementary Fig. 4g). Consistently, the targets of which half lifetime increased more with *METTL1* downregulation were also more stabilized upon *IGF2BP1* and *IGF2BP3* knockdown (Supplementary Fig. 4h). These results altogether suggest that IGF2BP1 and IGF2BP3 could promote decay of m⁷G-marked transcripts, including the METTL1 targets.

## m⁷G regulates mRNA stability

Similar as m⁶A (Supplementary Fig. 5a), m⁷G modifications are generally located at longer exons (Supplementary Fig. 5b, left), and the modified transcripts, especially those with m⁷G at the 3′UTR, are greater in length than unmodified ones (Supplementary Fig. 5b, right). While m⁶A and m⁷G are similar in distribution when bound by IGF2BPs (Fig. 2e and Supplementary Fig. 3c), they acted oppositely in half lifetime regulation (Fig. 3a and Supplementary Fig. 4c), which made us wonder about potential crosstalk among these two modifications on the same gene transcript. Unsurprisingly, more than half of the m⁷G-marked transcripts are also modified by m⁶A (Supplementary Fig. 5c). However, we did not observe a correlation of methylation levels between m⁷G and m⁶A at the transcript level (Fig. 3e). When we examined their potential overlap at the peak level, about 15% of the m⁷G peaks are adjacent to m⁶A peaks (Supplementary Fig. 5d), and the ratio increased to around 25%, occupying ~15% of the overall m⁶A peaks, in the 3′ UTRs (Supplementary Fig. 5e). But when we use IGF2BPs binding sites as reference, the overlapping peaks between both modifications in the 3′ UTR are limited (Fig. 3f), suggesting little conflict on the regulation of these two modifications through IGF2BPs on the same transcript. We further generated a HepG2 cell line with stable downregulation of *METTL3*, the catalytic component of the m⁶A writer complex, and evaluated RNA half lifetime using RT-qPCR. We observed a consistent stabilization effect on representative target transcripts modified by m⁷G upon *IGF2BP3* knockdown, suggesting that m⁷G targets can be regulated with little impact from m⁶A modification (Supplementary Fig. 5f).

To distinguish the potential opposite regulation in transcript lifetime by these two modifications, we grouped the target transcripts based on their corresponding m⁷G or m⁶A methylation level, respectively. When comparing m⁶A targets without m⁷G modification to those with high m⁷G modification (from lighter green to darker green in Fig. 3g and Supplementary Fig. 5g), we observed a gradual shift in the half lifetime change from destabilization towards stabilization upon either *IGF2BP1* (Supplementary Fig. 5g) or *IGF2BP3* (Fig. 3g) knockdown. Such patterns are consistent with the different regulatory functions of the two modifications.

To better evaluate the regulation of the IGF2BP proteins on the two modifications, we respectively compared the lifetime changes of their direct targets with either m⁶A or m⁷G, or both modifications (Fig. 3h). The dually modified transcripts showed a similar half lifetime change as those with only m⁷G modification upon *IGF2BP3* knockdown but acted more similarly as m⁶A-only targets upon *IGF2BP1* knockdown, suggesting that IGF2BP3 might be more responsive to m⁷G modifications, while IGF2BP1 prefers m⁶A. Such difference is also consistent with our previous binding affinity results, implying that individual members of the IGF2BP family play varied roles in mRNA stability regulation based on their preference to certain modifications.

We then performed functional enrichment analysis to illustrate the potential differences of the two modifications in biological functions (Supplementary Fig. 5h). m⁶A-modified transcripts enriched GO terms including protein ubiquitination, cell division, and transcription regulation, while m⁷G-modified transcripts are closely related to functions regarding cell cycle, chromatin remodeling, and mRNA regulation. Interestingly, the genes with both modifications highlighted the pathways of splicing and chromatin organization, indicating that the dual modifications might allow more flexible regulations on the transcription and splicing.

## in vitro validation with dCas13b tethering systems

As we have found that IGF2BP1 and IGF2BP3 could promote the degradation of m⁷G-marked transcripts, we took advantage of the catalytically inactive RNA-targeting CRISPR-Cas13 systems[53–55] to site-specifically manipulate m⁷G methylation and IGF2BPs binding. The nuclease inactivated Cas13 system (dCas13) has been widely used to

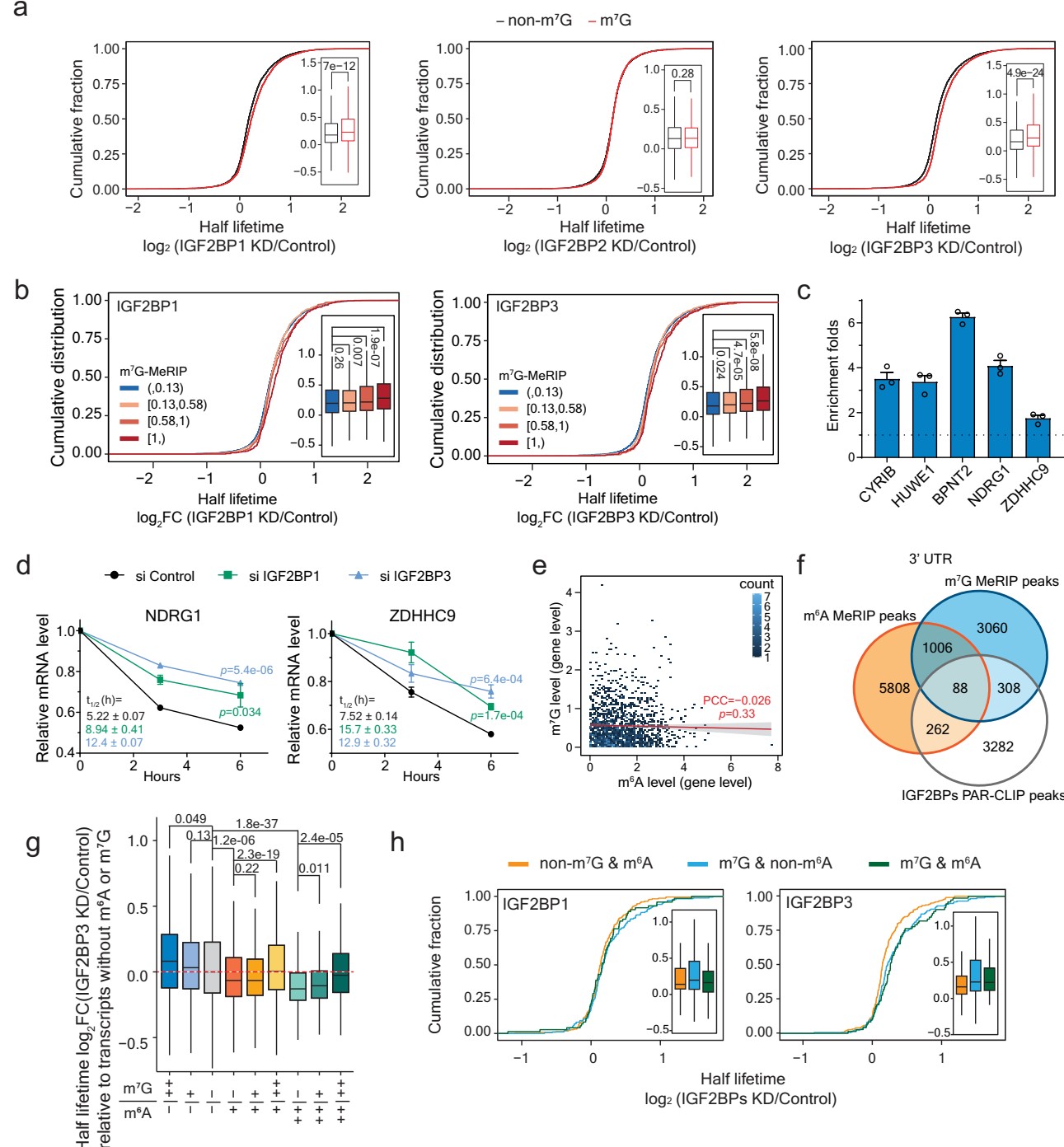

**Fig. 3 | IGF2BP1 and IGF2BP3 promote degradation of m⁷G-modified transcripts with limited interference from m⁶A. a** Cumulative curve of gene half lifetime fold-changes (log₂FC) upon *IGF2BP1* (left), *IGF2BP2* (middle), and *IGF2BP3* (right) knockdown in HepG2. Two independent replicates with two-sided Wilcoxon rank-sum test. Genes were categorized according to whether they were marked with m⁷G or not (non-m⁷G). The box plots on the right side extend from the first to the third quartile (Q1 to Q3), with lines in the middle representing median. Lines extending from both ends of boxes indicate variability outside the range. The minimum and maximum whisker values are calculated as Q1 − 1.5 * IQR and Q3 + 1.5 * IQR, respectively. **b** Cumulative curve of gene half lifetime fold-changes (log₂FC) upon *IGF2BP1* (top) and *IGF2BP3* (bottom) knockdown. Two independent replicates with two-sided Wilcoxon rank-sum test. Genes were categorized into four groups according to m⁷G enrichment level (log₂FC(IP/Input)). **c** Relative m⁷G methylation levels of the transcripts of the selected genes based on m⁷G-MeRIP-qPCR quantification. Mean ± SEM of three independent experiments. **d** Changes in mRNA levels in HepG2 cells with transient knockdown of *IGF2BP1* and *IGF2BP3*. Mean ± SEM

($n = 3$) with two-tailed Student's *t*-tests. Calculated half lifetimes marked in corresponding colors. **e** Pearson correlation between m⁷G levels and m⁶A levels on the transcripts of the same genes. Two-sided *p*-value with standard error marked in gray. **f** Venn diagram of the overlaps of m⁶A peaks, m⁷G peaks, and IGF2BPs PAR-CLIP peaks in 3′ UTR. **g** Boxplot showing gene half lifetime fold-changes (log₂FC) upon *IGF2BP3* knockdown. Two independent replicates with two-sided Wilcoxon rank-sum test. Genes were categorized into different groups according to their m⁷G or m⁶A modification level. '-' represents log₂FC(IP/Input) < 0; '+' represents log₂FC(IP/Input) > 0 and log₂FC(IP/Input) < 1; '++' represents log₂FC(IP/Input) > 1. **h** Cumulative curve of gene half lifetime fold-changes (log₂FC) upon *IGF2BP1* (left), and *IGF2BP3* (right) knockdown. Two independent replicates for each. Genes were categorized into three groups: (1) modified only by m⁶A (non-m⁷G & m⁶A), (2) modified only by m⁷G (non-m⁶A & m⁷G), and (3) modified by m⁷G and m⁶A (m⁷G & m⁶A). The box plots formatted as those in (**3a**). All source data are provided as a Source Data file.

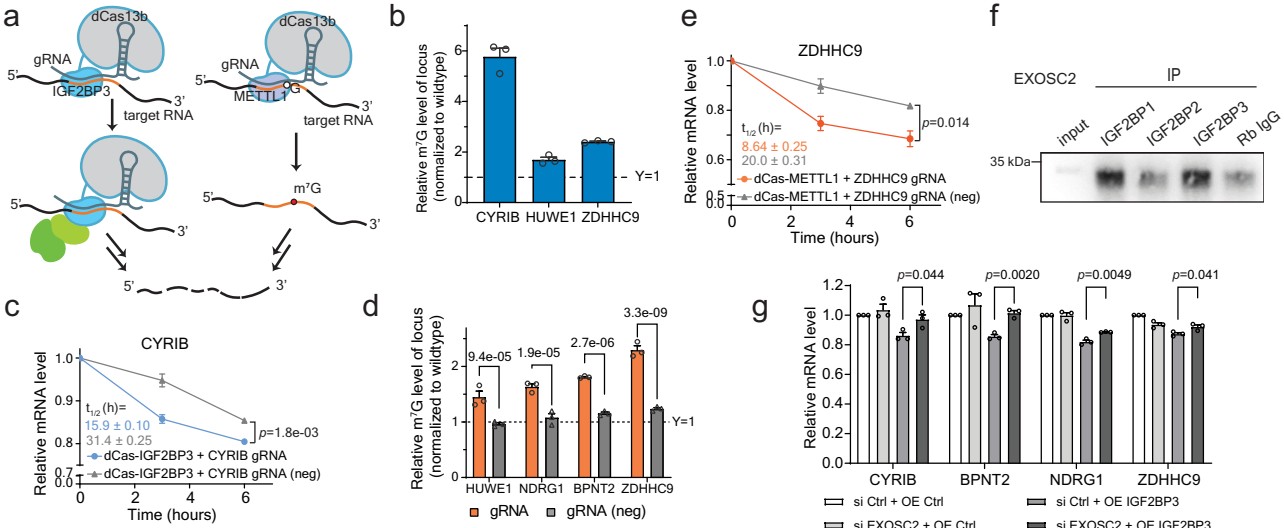

**Fig. 4 | Application of the dCas13b-tethering systems for cell-based validation and mechanistic studies. a** Schematic diagram showing the modified dCas13b system with IGF2BP3 (dCas13b-IGF2BP3, left) and METTL1 (dCas13b-METTL1, right). **b** Relative m$^7$G methylation levels of the selected gene loci. Mean ± SEM of three independent experiments. **c** Changes in mRNA levels in HepG2 cells with introduction of dCas13b-IGF2BP3 and guide RNA at the loci or not (neg). Mean ± SEM ($n = 3$) with two-tailed Student's $t$-tests. Calculated half lifetimes are marked in the corresponding colors. **d** Relative m$^7$G methylation level of the loci of the selected genes upon introducing the dCas13b-METTL1 tethering with the guide RNAs at the target loci or not (neg), normalized to the methylation level in the wildtype cells.

Mean ± SEM ($n = 3$) with two-tailed Student's $t$-tests. $P$-values are marked at the top of each group. **e** Changes in mRNA levels in HepG2 cells with the introduction of dCas13b-METTL1 and the guide RNAs at the loci or not (neg). Mean ± SEM ($n = 3$) with two-tailed Student's $t$-tests. Calculated half lifetimes are marked in the corresponding colors. **f** Western blotting of EXOSC2 in cell lysate from in vitro pulldown in HepG2 cells using the antibodies that recognize IGF2BP1, IGF2BP2, or IGF2BP3. Rabbit IgG was used as a negative control. This experiment was repeated independently twice with similar results. **g** mRNA levels of the m$^7$G targets upon IGF2BP3 overexpression, or *EXOSC2* knockdown, or both. Mean ± SEM ($n = 3$) with two-tailed Student's $t$-tests. All source data are provided as a Source Data file.

manipulate the methylation level, with either writer or eraser fused, and has also been used to investigate the reader binding effects on m$^6$A transcripts. We therefore fused the dCas13b system with IGF2BP3 as the reader representative or METTL1 as the writer (Fig. 4a) and tested whether these tethering constructs could affect the half lifetime of selected transcripts.

We picked several representative targets with methylation loci identified (Fig. 4b) and binding sites of IGF2BP proteins, especially IGF2BP3, near the methylation peaks (Supplementary Fig. 4e). These transcripts showed decreases in half lifetime when targeted by IGF2BP3 at the methylated loci rather than the unmethylated ones (Fig. 4c and Supplementary Fig. 6a). As a control, the tethering of IGF2BP1 presented little effect (Supplementary Fig. 6b), further indicating that IGF2BP3 is more responsive to m$^7$G while IGF2BP1 is not. We also directed dCas13b-METTL1 to the transcripts with relatively lowly methylated loci. The tethering of METTL1 increased the methylation levels of the targeted regions (Fig. 4d) and the binding intensity of IGF2BP3 on the transcripts based on the RNA immunoprecipitation (RIP) followed by qPCR quantification (Supplementary Fig. 6c). Such increases also resulted in elevated degradation of the transcripts (Fig. 4e and Supplementary Fig. 6d). The elevated degradation induced by tethering with IGF2BP3 or METTL1 could be greatly attenuated upon knockdown of *METTL1* or *IGF2BP3*, respectively (Supplementary Fig. 6e-f). We also applied the dCas13b system in cells with stable knockdown of *METTL3* and observed little difference compared to the control cells (Supplementary Fig. 6g-h). These results collectively suggest that both IGF2BP3 and m$^7$G modifications are indispensable for the regulation of transcript degradation, and such regulation is independent of m$^6$A.

## IGF2BP3 promotes the degradation of m$^7$G targets through exosome complex

We then asked how IGF2BP3 specifically promotes the degradation of the m$^7$G-modified transcripts. IGF2BP3 has been reported to interact

with XRN2 (5′−3′ exonuclease) and EXOSC2 (component of exosome complex) to promote the decay of the *eIF4E-BP2* mRNA[56]. We therefore asked whether these two proteins are also involved in the m$^7$G-dependent mRNA decay through IGF2BP3. Both XRN2 and EXOSC2 were enriched by the immunoprecipitation of the IGF2BP proteins (Fig. 4f and Supplementary Fig. 7a). The former showed a higher interaction level with IGF2BP2 (Supplementary Fig. 7a), while EXOSC2 is more closely interacted with IGF2BP1 and IGF2BP3 (Fig. 4f). Exosome complex is involved in many biology processes and EXOSC2 is known to located at the cap of the complex[57–59]. We next tested the association of IGF2BP proteins with other exosome complex components that are closely connected to EXOSC2. We found that when bound by EXOSC2, IGF2BP1 and IGF2BP3 form a close interaction with EXOSC4, EXOSC7, and EXOSC3, components close in location with EXOSC2 (Supplementary Fig. 7b). These results suggested that m$^7$G targets bound by IGF2BPs might be recognized and delivered to exosome complex through their interaction with EXOSC2 for accelerated degradation. To verify this, we knocked down *EXOSC2* in IGF2BP3 overexpressed HepG2 cells (Supplementary Fig. 7c) and observed that the decreased mRNA expression levels upon IGF2BP3 overexpression could be recovered by EXOSC2 downregulation (Fig. 4g), supporting the involvement of EXOSC2 and exosome in the IGF2BP3-mediated transcript decay.

## IGF2BP3 is involved in glioma and regulates the decay of *TP53* transcripts

As we have identified IGF2BP family as the binding proteins of internal m$^7$G and can promote decay of the target transcripts, we asked if IGF2BPs, especially IGF2BP3, are involved in the regulation of m$^7$G methylation in glioma given the correlation with METTL1 (Fig. 1). When we performed the gene ontology (GO) enrichment analysis with the transcripts targeted by IGF2BPs but only modified by m$^7$G, we noticed that the glioma pathway and the terms about cell cycle arrest and p53 pathways were both highlighted (Supplementary Fig. 8a), so as other

pathways closely related to p53, implying that IGF2BPs affect p53-related genes, especially in glioma, and this process might go through regulation on m[7]G-modified transcripts. IGF2BP3 is also highly correlated with the glioma term (Fig. 5a and Supplementary Fig. 8b).

Similar to METTL1, IGF2BP3 is highly expressed in tumors compared to normal tissues across various cancer types, with glioblastoma ranked second (Supplementary Fig. 8c). Notably, among the proteins involved in m[7]G regulation in glioblastoma, IGF2BP3 is the most upregulated one in tumor versus normal tissues (Supplementary Fig. 8d). In addition, higher expression of IGF2BP3 is also correlated to lower overall survival rate in glioma/glioblastoma (Supplementary Fig. 8e). Such correlation on the survival rate also dramatically diminished in patients with mutated p53 compared to those with wildtype p53 (Fig. 5b). All these correlations between IGF2BP3 with glioma and glioblastoma as well as the oncogenic functions of METTL1 in these tumors indicate that METTL1 might act as a critical player in tumorigenesis not only through m[7]G-modified tRNAs as previously reported[27,28], but also through internal m[7]G modifications within mRNAs and the mRNA m[7]G regulation by IGF2BP3.

We first confirmed that the methylated peak at the 3'UTR of *TP53* overlaps well with the IGF2BP3 binding (Supplementary Fig. 9a). In HepG2 cells, where the 3' UTR of *TP53* is moderately methylated, we knocked down *IGF2BP3* and observed a slower *TP53* decay (Supplementary Fig. 9b). Based on our previous quantification of the methylation level, we grouped the four glioblastoma cell lines into highly (U87MG and LN229) and lowly (A172 and T98G) m[7]G-methylated groups. All cell lines presented significant increases in the half lifetime of *TP53* transcript upon *IGF2BP3* knockdown (Fig. 5c and Supplementary Fig. 9c). We then performed knockdown of *METTL1* in the highly methylated group. We observed a consistent decrease of the m[7]G level in the 3' UTR of *TP53* (Fig. 5d), which confirmed *TP53* as an METTL1 target. Also, in the highly methylated group, the overexpression of IGF2BP3 led to a dramatic decrease in the methylation level of *TP53* at its 3' UTR locus, especially in LN229 (Fig. 5e), where the IGF2BP3 protein is limited (Supplementary Fig. 9d). The overexpression of IGF2BP3 would specifically promote the degradation of m[7]G targets compared to the non-methylated ones, and thus decrease the relative methylation level of *TP53* at its 3' UTR region. Correspondingly, in the lowly methylated group, *IGF2BP3* knockdown resulted in increased *TP53* m[7]G methylation levels (Fig. 5f).

As cells in the highly methylated group are more sensitive to IGF2BP3 overexpression, we further validated our decay mechanism in these cell lines (Supplementary Fig. 9e). As expected, the transcript level of *TP53* was recovered upon *EXOSC2* knockdown in IGF2BP3 overexpressed cells (Supplementary Fig. 9f), supporting that IGF2BP3 tunes *TP53* stability through the EXOSC2-mediated pathway. These results implied that in glioblastoma cells, IGF2BP3 could regulate the *TP53* transcript level through its recognition and action on the m[7]G at its 3' UTR. To establish a causal relationship, we directed dCas13b-fusion systems to the *TP53* 3' UTR locus in glioblastoma. The transcript targeted by IGF2BP3 was more rapidly decayed (Fig. 5g, left, and Supplementary Fig. 9g, left), which is consistently observed across the tested cell lines. In the cells with lowly methylated 3' UTR, the tethering of METTL1 increased the methylation level in the locus of *TP53* (Fig. 5h), which also led to a more rapid degradation of *TP53* (Fig. 5g, right, and Supplementary Fig. 9g, right).

### Downregulation of *TP53* by m[7]G and IGF2BP3 affects cell proliferation and chemoresistance

The introduction of dCas13b systems to *TP53* in the glioblastoma cells, fused with either IGF2BP3 or METTL1, not only suppressed the transcript level (Fig. 5i), but also led to a decreased p53 protein level (Fig. 5j and Supplementary Fig. 9h), with dCas13b-IGF2BP3 showing more decreases. As p53 is widely involved in cancer regulation, we further validated the effects of the downregulated *TP53* on related pathways.

The wildtype p53 plays an important role in cell cycle regulation. Considered as the guardian of genome, p53 is highly involved in DNA damage repair, cell cycle regulation, and apoptosis[60–62]. The p53 pathway has previously shown to be regulated through mRNA modifications such as m[6]A[63]. In LN229 cells with partially functional p53[38], or U87MG cells with wild-type p53, the downregulation of p53 level triggered by the dCas13b-IGF2BP3 tethering system could promote cell proliferation (Fig. 5k and Supplementary Fig. 9i), in a similar way to the knockdown of *TP53* using siRNA transfection (Supplementary Fig. 9j) or overexpression of IGF2BP3 (Supplementary Fig. 9k). The decreased protein level of p53 upon dCas13b-IGF2BP3 tethering also led to a decreased percentage of G0/G1 cells, suggesting an increased level of cells escaping G1 arrest (Supplementary Fig. 9l-m). Such results are consistent with the role of p53 for arrest induction in regulation cell cycle checkpoint.

Besides cells with wildtype p53, we also investigated the gain of function p53 mutant in the T98G cells. T98G is a well-studied TMZ resistant cells with IC50 usually over 500–1000 μM[64–66]. The mutant p53 in T98G cells elevated the expression level of O[6]-methylguanine-DNA methyltransferase (MGMT), a key regulator in DNA damage repair, and thus promoted TMZ resistance[67]. We picked the dCas13b-IGF2BP3 tethering system where p53 proteins were more downregulated and validated that both the mRNA level (Supplementary Fig. 9n) and protein expression (Fig. 5l) of MGMT were decreased upon introduction of dCas13b-IGF2BP3 with the guide RNA. We then continued with the TMZ resistance test. The wildtype cells presented a generally high resistance towards TMZ treatment. However, T98G cells introduced with dCas13b-IGF2BP3 and the guide RNA demonstrated a noticeable decrease in TMZ resistance (Fig. 5m), which is also observed upon IGF2BP3 overexpression (Supplementary Fig. 9o), confirming a regulation of p53 function through the internal m[7]G methylation.

Altogether, our results revealed that the IGF2BP family proteins could recognize mRNA internal m[7]G modification. The binding of IGF2BP3 and IGF2BP1 could promote the decay of m[7]G-modified transcripts, with IGF2BP3 exhibiting a higher preference to m[7]G and more effects on target transcript decay. In glioblastoma, we show that the tuning of m[7]G methylation in the 3' UTR of *TP53* could manipulate the mRNA and protein levels of p53, which affected cancer cell proliferation or chemoresistance.

## Discussion

The internal m[7]G modification has been found to impact biological processes through tRNA and rRNA. While the single-base resolution maps of internal m[7]G at the whole transcriptome level are available, it remains underexplored whether the positive-charged mRNA modification plays any role in the mRNA regulation and the underlying pathways. Here, we identified the IGF2BP family proteins as the binding proteins of the mRNA internal m[7]G in cancer cells. The recognitions of m[7]G by IGF2BP1 and IGF2BP3 promote the degradation of the methylated transcripts, with IGF2BP3 more responsive to the modification. We found that *TP53* is m[7]G-modified in its 3' UTR with varied methylation levels across glioblastoma cell lines; the methylation is also negatively correlated with transcript expression level across these glioblastoma cell lines, suggesting a destabilization effect. The tethering of METTL1 or IGF2BP3 to *TP53* led to downregulation of *TP53* expression, respectively, leading to changes of cell proliferation and chemoresistance.

Both METTL1 and IGF2BP3 are highly expressed in glioblastoma, and their upregulations are associated with poor survival in patients. The correlation was more obvious in patients with wildtype p53 but diminished in the mutant ones. Here, based on our results, the upregulation of both METTL1 and IGF2BP3 could lead to degradation of *TP53* transcripts and thus decreased p53 protein level. The downregulation of functional wildtype p53 would promote tumor progression. *TP53* is one of the most frequently mutated genes in tumors and

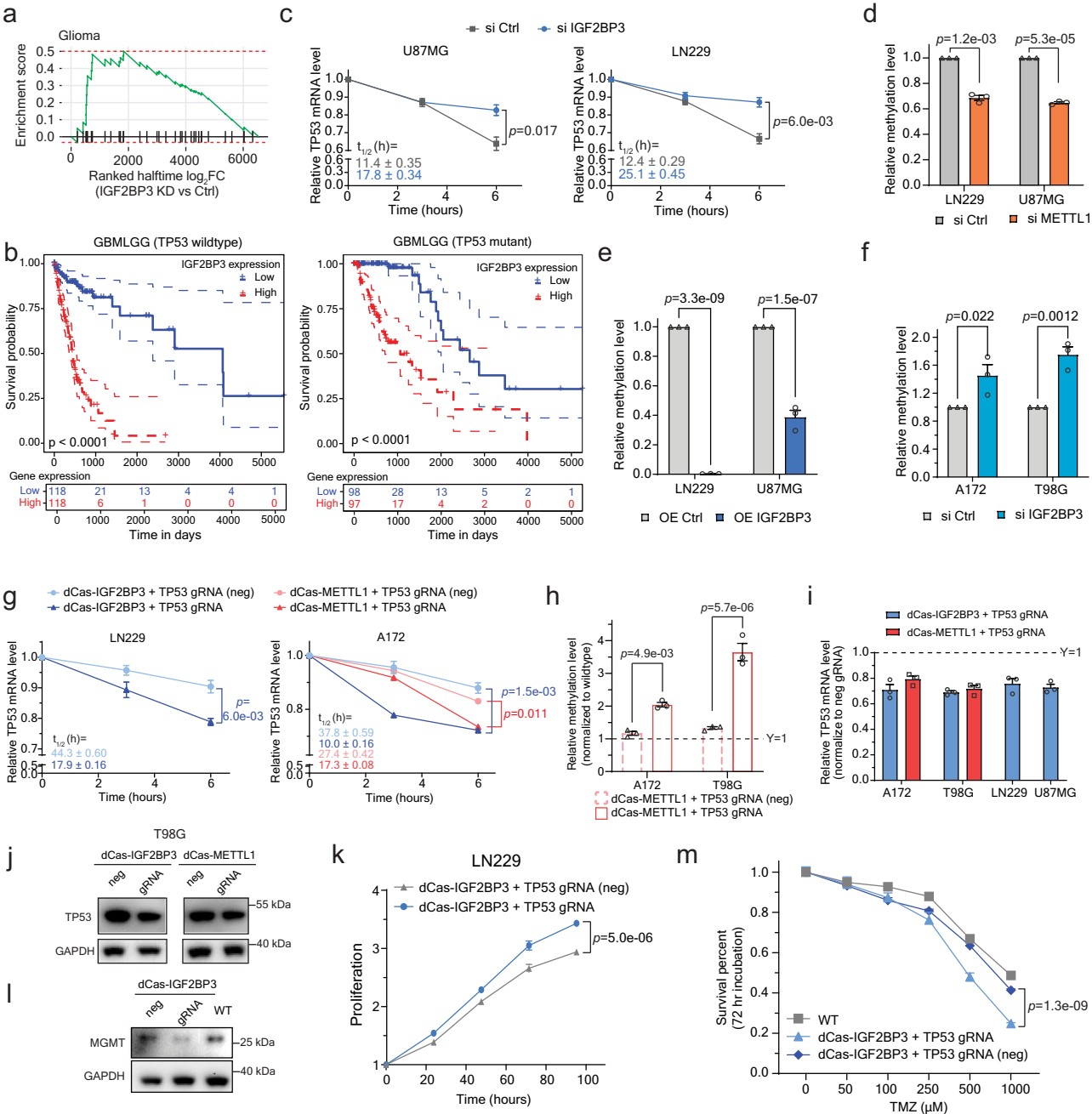

**Fig. 5 | Regulation of IGF2BP3 on m7G-marked transcripts in glioblastoma cells.**
**a** Gene Set Enrichment Analysis (GSEA) of genes in 'Glioma' KEGG pathway against ranked list of genes according to half lifetime changes upon *IGF2BP3* knockdown. **b** Kaplan-Meier survival analysis in TCGA database for Glioma (TCGA-GBMLGG) with wildtype *TP53* (left), or with mutant *TP53* (right). The patients were divided into two groups of equal size based on *IGF2BP3* levels. *P*-value detected by log-rank test. **c** Changes in *TP53* mRNA level in highly methylated cells upon *IGF2BP3* knockdown. Mean ± SEM (*n* = 3) with two-tailed Student's *t*-tests. Calculated half lifetimes marked in corresponding colors. **d**–**f** Relative m7G levels of the *TP53* 3' UTR locus upon *METTL1* knockdown (**d**), IGF2BP3 overexpression (**e**), and *IGF2BP3* knockdown (**f**), normalized to the methylation level in the corresponding control cells. Mean ± SEM (*n* = 3) with two-tailed Student's *t*-tests. **g** Changes in mRNA levels in cells with introduction of dCas13b-IGF2BP3 or dCas13b-METTL1 and *TP53* gRNA at the loci or not (neg). Mean ± SEM (*n* = 3) with two-tailed Student's *t*-tests. Calculated half lifetimes marked in corresponding colors. **h** Relative m7G levels of the *TP53* 3'

UTR locus with introduction of dCas13b-IGF2BP3 or dCas13b-METTL1 and *TP53* gRNA at the loci or not (neg), normalized to the methylation level in wildtype cells. Mean ± SEM (*n* = 3) with two-tailed Student's *t*-tests. **i** Relative *TP53* mRNA levels in cells with dCas13b-IGF2BP3 or dCas13b-METTL1 and the guide RNA, normalized to the cells with negative gRNA. Mean ± SEM (*n* = 3) with two-tailed Student's *t*-tests. **j** Western blot of p53 levels in T98G cells with dCas13b-IGF2BP3 or dCas13b-METTL1 tethering with the guide RNA or the negative gRNA. **k** Cell proliferation in LN229 cells with dCas13b-IGF2BP3 tethering with the guide RNA or the negative gRNA. Mean ± SEM (*n* = 6) with two-tailed Student's *t*-tests. **l** Western blot of MGMT levels in T98G cells with dCas13b-IGF2BP3 tethering with the guide RNA compared to negative gRNA and wildtype cells. **m** Survival percentages of T98G cells treated TMZ at different concentrations with dCas13b-IGF2BP3 tethering with the guide RNA, or the negative gRNA, compared to wildtype cells. Mean ± SEM (*n* = 6) with two-tailed Student's *t*-tests. **j** and (**l**) were both repeated independently twice with similar results. All source data are provided as a Source Data file.

can protect cancer cells from oxidative and proteotoxic stresses. Loss-of-function or gain-of-function mutants could lead to diverse effects in different tumors. We also included our study in T98G cells with a gain-of-function *TP53* where the m⁷G methylation of the mutant *TP53* transcript increased its chemosensitivity to temozolomide, showcasing consistent transcript decay effect of m⁷G and IGF2BP3 in cancer lines with diverse *TP53* status. We also performed active mRNA translation sequencing (ART-seq) to capture actively translating transcripts upon *IGF2BP3* knockdown (Supplementary Fig. 10a) and validated the protein expression level changes of the representative targets in both HepG2 (Supplementary Fig. 10b and 10c) and other cancer cell lines (Supplementary Fig. 10d and 10e). While *IGF2BP3* knockdown impaired the translation of m⁶A targets as previously reported[46], IGF2BP3 depletion generally elevated transcript levels of m⁷G targets but presented limited impact on their translation, further suggesting its regulatory role on m⁷G targets in mRNA decay.

Human IGF2BP proteins share about 56% overall sequence identity and the identity increases to around 70% when only considering the RNA binding domains[47]. Though they share similar amino acid sequences, they differed in the RNA modification recognition as revealed in our study. While IGF2BP2 mostly recognizes m⁶A, IGF2BP1 and IGF2BP3, particularly IGF2BP3, could promote decay of the m⁷G-modified target transcripts. Although differences in binding motif preferences may partially account for their binding diversity, detailed elucidation of their structural alterations and potential protein interactions upon binding with distinct RNA modifications will allow a comprehensive understanding of diverse regulatory roles of IGF2BP proteins. IGF2BP1 and IGF2BP3 are more homologous to each other and display limited expression levels in adult organs except for the reproductive tissues. On the contrary, IGF2BP2 is widely expressed in various adult tissues. These all suggest that IGF2BP family proteins could play diverse roles on mRNA based on their distinct expressions and binding preferences. These regulations could be cell context dependent. Certain cancer cells may possess higher abundances of mRNA internal m⁷G and thus be subjected to stronger IGF2BP3 regulation through m⁷G.

While all IGF2BP family members recognize both m⁷G and m⁶A modifications, we observed an opposite trend for regulation on mRNA degradation where m⁷G seems to add another layer of regulation. The dynamics of transcripts bearing these modifications may vary, even when they are recognized by the same protein, due to changes in the abundance or activity of these regulatory factors. Alterations in local motifs and structural arrangements near a modified site may also readily modulate its access to specific reader proteins. Additionally, the localization of binding sites, whether within coding sequence (CDS) or untranslated regions (UTRs), could influence the regulatory pathways. Interestingly, our analysis revealed that while the interactions between IGF2BPs and m⁶A or m⁷G both predominantly occur in the 3′UTR, the overlapped peaks of IGF2BPs and m⁶A are mainly concentrated near the stop codon. Conversely, interactions with m⁷G span across the entire 3′UTR, implying that IGF2BP proteins might bind to different modifications at distinct regions within the 3′UTR. This potentially could lead to recruitment of different partners for RNA regulation, likely in different cell types and cell contexts.

In addition to the decay function shown in this study, previous reports discovered that internal m⁷G, especially those as METTL1 targets, could promote translation of modified transcripts. Unlike m⁶A, m⁷G tends to enrich in the CDS region[21]. While IGF2BPs mainly bind to 3′ UTR and regulate mRNA decay, the m⁷G in the CDS might act in a different way through interaction with ribosome and affect translation. METTL1 may only account for a subset of the internal m⁷G installation, other writer(s) might also be involved and regulate certain targets. Deciphering the regulators and the functions of internal m⁷G should allow better understanding of the overall orchestration of mRNA

regulation pathways through different modifications in mammalian processes and disease development.

## Methods

### Cells culture

Human HepG2, A172, LN229, T98G, and U87MG cell lines used in this study were all purchased from ATCC (the American Type Culture Collection). HepG2 and A172 cell lines were grown in DMEM (Gibco, 11995) media supplemented with 10% FBS and 1% 100X Pen/Strep (Gibco). LN229 cell line was grown in DMEM (Gibco, 11995) media supplemented with 5% FBS and 1% 100X Pen/Strep (Gibco). T98G and U87MG cell lines were maintained in EMEM (ATCC, 30-2003), supplemented with 10% FBS and 1% 100X Pen/Strep (Gibco). All cells were cultured at 37 °C under 5.0% $CO_2$. To construct the METTL3 knockdown and control cell lines, we used the TRC Lentiviral Human shRNA system encoding a control shRNA or a shRNAs targeting METTL3 (TRCN0000034715).

### Antibodies

The antibodies used in this study are listed below in the format of name (catalog; supplier; dilution fold): Mouse anti-m⁷G (RN017M; MBL; 1000; clone 4141-13). Rabbit anti-METTL1 (14994-1-AP; Proteintech; 1000). Rabbit anti-m⁶A (E1610S; NEB; 1000). Mouse anti-WDR4 (sc-100894; Santa Cruz; 100). Rabbit anti-IGF2BP1 (8482; Cell Signaling; 1000). Rabbit anti-IGF2BP2 (14672; Cell Signaling; 1000). Rabbit anti-IGF2BP3 (57145; Cell Signaling; 1000). Rabbit anti-EXOSC2 (ab181211; Abcam; 10000). Mouse anti-EXOSC3 (sc-166568; Santa Cruz; 100). Mouse anti-EXOSC4 (sc-166772; Santa Cruz; 100). Mouse anti-EXOSC7 (sc-393686; Santa Cruz; 100). Rabbit anti-XRN2 (13760; Cell Signaling; 1000). Mouse anti-p53 (sc-126; Santa Cruz; 100). Mouse anti-MGMT (sc-166528; Santa Cruz; 100). Mouse Anti-BrdU (B2531; Sigma; 100; clone BU-33). Goat Anti-Mouse IgG H&L (Alexa Fluor® 488) (ab150113; Abcam; 2000). Goat anti-rabbit IgG-HRP (7074; Cell Signaling; 3000). Horse anti-mouse IgG-HRP (7076; Cell Signaling; 3000). Rabbit anti-GAPDH-HRP (8884; Cell Signaling; 1000).

### RNA isolation

Generally, to isolate total RNA from cells, the media was aspirated, and the cells were washed once with a proper volume of ice-cold DPBS buffer for each plate. Then total RNA was isolated from cells with TRIzol reagent (Invitrogen) and then extracted following the manufacturer's protocol through isopropanol precipitation. GlycoBlue Coprecipitant (15 mg ml⁻¹, Thermo Fisher Scientific) was added to the solution during precipitation if needed. mRNA enrichment: starting from extracted total RNA, mRNA was purified with two rounds of polyA⁺ purification with Dynabeads mRNA DIRECT kit (Ambion). RNA concentration was measured by UV absorbance at 260 nm or using Qubit RNA HS Assay Kit (Thermo Fisher Scientific) with Qubit 2.0 fluorometer.

### LC-MS/MS

Around 200-300 ng mRNA was digested first with nuclease S1 (1uL, Sigma) in a 20 μL reaction buffer containing 10 mM of $NH_4OAc$ (pH = 5.3) at 42 °C for 2 h. Then, 1 μL of shrimp alkaline phosphatase (rSAP, NEB) was added along with 2.5 μL of 10X CutSmart buffer (NEB) and incubated at 37 °C for 2 h. After the incubation, the sample was diluted with additional 35 μL water and filtered with 0.22 μm filters (4 mm diameter, Millipore) and 8 μl of the entire solution was injected into LC-MS/MS as one sample. For all the quantification, a mock control with only the digestion buffers and enzymes and water was included each time. The signals from the mock control would be later subtracted from the signals from experimental samples as the baseline. Nucleosides were separated, by reverse-phase ultra-performance liquid chromatography, on a C1 column with on-line mass spectrometry detection by an Agilent 6410 QQQ triple-quadrupole LC mass

spectrometer, in positive electrospray ionization mode. The nucleosides were quantified with retention time and the nucleoside-to-base ion mass transition of 284-152 (G), 268-136 (A), 298.1-166.1 ($m^7G$). Quantification was performed in comparison with the standard curve, obtained from pure nucleoside standards running with the same batch of samples. The $m^7G$ level was calculated as the ratio of $m^7G$ to G based on calibrated concentration curves.

## Decapping of mRNA

Decapping of mRNA was performed with Tobacco Decapping Plus 2 (#94, Enzymax). The reaction was prepared, with a maximum of 6 µg fragmented mRNA in nuclease-free water with 5 µL 10X Decapping Reaction Buffer (100 mM Tris-HCl pH 7.5, 1.0 M NaCl, 20 mM $MgCl_2$, 10 mM DTT), 1uL 50 mM $MnCl_2$, 2.5 µL SUPERase-In RNase Inhibitor (Thermo Fisher Scientific) and 8 µL Tobacco Decapping Plus 2 enzymes, diluted to a final volume of 50 µL. The reaction was incubated at 37 °C for 2 h. Decapped RNA was extracted from the solution with RNA Clean & Concentrator (Zymo Research).

## Western blot

Samples were homogenized in CelLytic M buffer (Sigma) containing 1 × protease inhibitor cocktail (Roche) on ice for at least 15 min. The lysates were then centrifuged to remove the cellular debris and boiled at 95 °C with 4 × loading buffer (Bio-Rad) for 5 min and stored at −80 °C for later use in the next step. 5 µg or more total protein amount per sample was loaded into 4–12% NuPAGE Bis-Tris gel (Life Technologies) and transferred to nitrocellulose membranes (Bio-Rad). Membranes were blocked in 5% milk in TBST for 30 min at room temperature, incubated in a diluted primary antibody solution at 4 °C overnight, washed and incubated in a dilution of secondary antibody conjugated to HRP for 1 h at room temperature. Protein bands were detected with SuperSignal West Dura Extended Duration Substrate kit (Thermo) or SuperSignal West Femto Maximum Sensitivity Substrate (Thermo) if needed and FluroChem R (Proteinsimple).

## EMSA (electrophoretic mobility shift assay/gel shift assay)

The RNA probes were synthesized with MEGAshortscript T7 Transcription Kit (Thermo) based on the sequence of 5'- CCAATAAAA-TATTAACCACCAATAAAATATTAACCAA**Z**ATCCACCAATAAAA-TATTAACC-3' (**Z** = G or $m^7G$) with either GTP (included in the kit) or $m^7GTP$ (Sigma, dissolve as 75 mM). After the synthesis, the RNA probe was labeled with T4 RNA ligase 1 (NEB) and pCp-Cy3 (Jena Bioscience), following the manufacturer's protocol. The RNA probes were denatured at 65 °C for 4 min, and then quickly cooled on ice. FLAG–IGF2BP1, FLAG–IGF2BP2, and FLAG–IGF2BP3 were purified with overexpression in HEK293T cells followed by FLAG-tag pulldown with anti-Flag M2 magnetic beads (Sigma) following commercial protocol. The protein purifications were validated with 4-12% NuPAGE Bis-Tris gel running followed by protein staining with Imperial Protein Stain (Thermo). The proteins were diluted to the desired concentration series in binding buffer (10 mM HEPES, pH 8.0, 50 mM KCl, 1 mM EDTA, 0.05% Triton-X-100, 5% glycerol, 10 µg ml⁻¹ salmon DNA, 1 mM DTT and 40 U ml⁻¹ RNasin). Before loading to each well, 1 µl RNA probe (4 nM final concentration) and 1 µl protein (20 nM, 50 nM, 100 nM, 200 nM, 300 nM or 1 µM final concentration) were added and the solution was incubated on ice for 30 min. The entire 10 µl RNA–protein mixture was loaded to the Novex 4-20% TBE gel (Invitrogen) and run at 4 °C for 90 min at 90 V. Imaging was carried out by Bio-Rad Molecular Imager FX under Cy3 channel.

## in vitro pulldown for western and protein mass spectrometry

Probes were prepared through in vitro transcription as described before. After the synthesis and cleanup with RNA Clean and Concentrator (Zymo Research), the RNA probes were labeled by Pierce

RNA 3' End Biotinylation Kit (Thermo) followed by purification with RNA Clean and Concentrator (Zymo Research). The $m^6A$ and A probes (5'- GAACCGG**X**CUGUCUUA-3' (**X** = A or $m^6A$)) were synthesized directly with biotin tag at 5' end. HepG2 cells were collected (one 15-cm plate) by cell lifter (Corning Incorporated), pelleted by centrifuge for 5 min at 500 g and washed once with cold PBS (6 ml). The cell pellet was re-suspended with 2 volumes of lysis buffer (250 mM NaCl, 10 mM Tris-HCl pH 7.6, 0.5% NP-40, 10% glycerol, 1:100 protease inhibitor cocktail, 400 U ml⁻¹ RNase inhibitor) and mixed with rotation at 4 °C for 30 min. The mixture was then centrifuged at 15,000 g for 15 min to clear the lysate. 50 µl cell lysate was saved as input. The rest was divided into two tubes with equal volume. 1 µg G or $m^7G$ probes were added to the tubes. The probe and cell lysate were then rotated continuously at 4 °C for 2 h. 20 µl Dynabeads MyOne Streptavidin C1 beads (Invitrogen) were washed for each sample with the lysis buffer and then added to the mixture, and underwent another 2 h incubation at 4 °C. The beads were collected and washed five times with 1 ml ice-cold lysis buffer. The beads were sent directly to MS Bioworks for protein mass spectrometry analysis or boiled with 40 µl 1X loading buffer (Bio-Rad) (diluted with PBS) at 95 °C for 5 min, followed by western blot analyses.

For protein mass spectrometry, two samples were processed, one for G probe as control, and the other for $m^7G$ probe. The samples were eluted in reducing LDS buffer at 100 °C for 15 min. Half of each sample was processed by SDS-PAGE using 10% Bis-Tris NuPage Mini-gel (Invitrogen) with the MES buffer system. The gel was run 1 cm and the mobility region excised into 10 equally sized bands. Each band was processed by in-gel digestion with trypsin using a robot (ProGest, DigiLab) with the following protocol: (1) washed with 25 mM ammonium bicarbonate followed by acetonitrile; (2) reduced with 10 mM dithiothreitol at 60 °C followed by alkylation with 50 mM iodoacetamide at room temperature; (3) digested with sequencing grade trypsin (Promega) at 37 °C for 4 h; (4) Quenched with formic acid and the supernatant was analyzed directly without further processing. Half of each digested sample was analyzed by nano LC-MS/MS with a Waters NanoAcquity HPLC system interfaced to a ThermoFisher Fusion Lumos mass spectrometer. Peptides were loaded on a trapping column and eluted over a 75 µm analytical column at 350 nL/min; both columns were packed with Luna C18 resin (Phenomenex). The mass spectrometer was operated in data-dependent mode, with the Orbitrap operating at 60,000 FWHM and 15,000 FWHM for MS and MS/MS respectively. The instrument was run with a 3 s cycle for MS and MS/MS. 5-h instrument time was used for the analysis of each sample. Data were searched using a local copy of Mascot (Matrix Science) with the following parameters: Enzyme: Trypsin/P; Databases: SwissProt Human (concatenated forward and reverse plus common contaminants); Fixed modifications: Carbamidomethyl (C); Variable modifications: Acetyl (N-term), Deamidation (N,Q), Oxidation (M), Pyro-Glu (N-term Q); Mass values: Monoisotopic; Peptide Mass Tolerance: 10 ppm; Fragment Mass Tolerance: 0.02 Da; Max Missed Cleavages: 2. Mascot DAT files were parsed into Scaffold (Proteome Software) for validation and filtered to create a non-redundant list per sample. Data were filtered at 1% protein and peptide FDR. At least two unique peptides are required per protein.

## Protein coimmunoprecipitation

HepG2 cells were collected by cell lifter (one 15 cm plate), and pelleted by centrifuge at 500 g for 5 min. The cell pellet was resuspended with 2 volumes of lysis buffer (150 mM KCl, 10 mM HEPES pH 7.6, 2 mM EDTA, 0.5% NP-40, 0.5 mM DTT, 1:100 protease inhibitor cocktail, 400 U ml⁻¹ RNase inhibitor), and incubated on ice for 30 min. To remove the cell debris, the lysate solution was centrifuged at 15,000 g for 15 min at 4 °C. While 50 µl of cell lysate was saved as the input, the rest was incubated with the anti-IGF2BP1 or anti-IGF2BP3 antibodies for 2 h at

4 °C. Afterwards, Protein A beads (Invitrogen) were washed and added to the mixtures with another 2 h incubation at 4 °C. The beads were then washed with ice-cold NT2 buffer (200 mM NaCl, 50 mM HEPES pH 7.6, 2 mM EDTA, 0.05% NP-40, 0.5 mM DTT, 200 U ml⁻¹ RNase inhibitor) four times. The eluted samples, saved as IP, were analyzed along with the input samples by western blotting.

## Quantification of RNA methylation with m⁷G-IP and RT-qPCR

We performed m⁷G-MeRIP enrichment followed by RT-qPCR to quantify the relative m⁷G methylation level or level changes of certain target m⁷G sites or the entire transcripts. 0.5 μg purified polyA⁺ RNA extracted from the cells of interest were fragmented (or not if using full length mRNA) with Bioruptor Pico (Diagenode) sonication with 30 s ON/ 30 s OFF for 8 cycles. 1 μL 1:100 diluted non-modified spike-in from EpiMark $N^6$-Methyladenosine Enrichment Kit (NEB) was added to each sample, and m⁷G-MeRIP was performed with the methylation-specific antibody (MBL). 2 μL anti-m⁷G antibody (MBL) in 250 μL 1X IPP buffer (10 mM Tris-Cl, pH = 7.4; 150 mM NaCl; 0.1% NP-40) with freshly added 5% SUPERase-In RNase inhibitor (Thermo Fisher Scientific) at 4 °C for 2–4 h. Then 20 μL Dynabeads Protein G resins (Thermo Fisher Scientific) were washed twice with 1X IPP buffer, resuspended in 20 μL IPP buffer, and added to the antibody-RNA mixture for another 2 h at 4 °C. The resins were then washed with 1X IPP buffer at 4 °C four times. RNA was finally eluted with Proteinase K (recombinant, PCR grade, EO0491, Thermo Fisher Scientific) digestion. 45 μL 1X Proteinase K digestion buffer (2X recipe: 2% SDS, 12.5 mM EDTA, 100 mM Tris-Cl (pH = 7.4), 150 mM NaCl) with 5 μL Proteinase K was used to resuspend the resins, and the solution was incubated at 55 °C for 30 min. The m⁷G-containing RNA was recovered with RNA Clean & Concentrator (Zymo Research), reverse transcribed with PrimeScript RT Master Mix (Takara), and then subjected to RT-qPCR. The spike-in was used as a reference gene when performing qPCR.

## RNA-seq for mRNA lifetime

Five 10-cm plates of HepG2 cells were transfected with IGF2BP1-3 siRNA or METTL1 siRNA or control siRNA at 30% confluency. After 6 h, each 10-cm plate was re-seeded into three 6-cm plates, and each plate was controlled to afford the close numbers of cells. After 48 h, actinomycin D was added to 5 μg ml⁻¹ at 6 h, 3 h, and 0 h before collection. The total RNA was purified with TRIzol (Invitrogen) described before. ERCC RNA spike-in control (Ambion) was added to each sample (0.02 μl per 1 μg total RNA). The total RNA with spike-in controls were then purified to acquire mRNA and the libraries were constructed with SMARTer Stranded Total RNA-Seq Kit v2 (Takara) according to the manufacturer's protocols and subject to Illumina Nova Seq in single-end mode with 100 base pair per read.

## RNA lifetime measurement by qPCR

Samples were prepared as RNA-seq for mRNA lifetime. Total RNAs were extracted from all the samples and non-modified spike-in from EpiMark $N^6$-Methyladenosine Enrichment Kit (NEB) was added if external control is needed. Total RNAs were reverse transcribed with PrimeScript RT Master Mix (Takara), and then subjected to RT-qPCR. The spike-in was used as a reference gene when performing qPCR.

## RT-qPCR

Quantitative reverse transcription PCR (RT-qPCR) was used to assess the relative abundance of RNA. Total RNA or purified nonribosomal RNA was reverse transcribed with PrimeScript RT Master Mix (Takara) to obtain cDNA. qPCR was performed by using FastStart Essential DNA Green Master (Roche) in machine LightCycler 96 (Roche). GAPDH were used as internal controls in different cases. When external control needed, 1 μL 1/50-1/200 diluted non-m⁶A spike-in from EpiMark $N^6$-Methyladenosine Enrichment Kit were added to each sample.

## dCas13b-IGF2BP3 and METTL1 reporter assay

dCas13b plasmid was a gift from Dr. Bryan Dickson (University of Chicago). dCas13b-IGF2BP3 and dCas13b-METTL1 were generated accordingly. The plasmids were sequenced by the University of Chicago Comprehensive Cancer Center DNA Sequencing and Genotyping Facility. For 6-well assays, cells were transfected with 1 μg dCas13b proteins and 1.5 μg gRNA for 24 hr before analysis.

## Cell proliferation assay

5000 cells were seeded per well in a 96-well plate. The cell proliferation was assessed by assaying cells at various time points using the CellTiter 96® Aqueous One Solution Cell Proliferation Assay (Promega) following the manufacturer's protocols. For each sample tested, the signal from the MTS assay was normalized to the value observed at ~5 or 24 h after seeding.

## Cell cycle analysis

LN229 and U87MG cells were labeled with BrdU (10 μM) for 30 min under normal incubator conditions, then trypsinized and fixed with 70% ethanol overnight. After fixation, cell pellets were collected, resuspended in 200 μL 4 M HCl, and incubated at room temperature for 20 min, followed by neutralization with Borax. The cell pellets were washed twice with 1% BSA in PBS before incubation with the BrdU primary antibody (B2531; Sigma; clone BU-33) at room temperature for 30 min in the dark. After three washes, the pellets were incubated with goat anti-mouse IgG Alexa Fluor 488 secondary antibody (ab150113; Abcam) at room temperature in the dark for 1 h. Cell pellets were then washed three times with 1 mL 0.1% Triton X-100 and 1% BSA in PBS before incubation with RNase A and propidium iodide (PI) at a final concentration of 5 μg/mL PI for 15–30 min at room temperature in the dark. Samples were analyzed using a BD LSR Fortessa flow cytometer, and the data were analyzed with FlowJo software (BD Biosciences) with the Cell Cycle model.

## siRNA knockdown and plasmid transfection

AllStars negative control siRNA from Qiagen (1027281) was used as control siRNA in knockdown experiments. *IGF2BP1-3*, *TP53*, *EXOSC2*, and *METTL1* siRNAs were ordered from Dharmacon as pre-designed. Transfection was achieved by using Lipofectamine RNAiMAX (Invitrogen) for siRNA, and Lipofectamine 2000 (Invitrogen) for single type of plasmid or Lipofectamine LTX Plus (Invitrogen) for co-transfection of two or multiple types of plasmids (tethering assay) following the manufacturer's protocols.

## PAR-CLIP

We followed the previously reported protocol. Eight 15 cm plates of HepG2 cells were seeded for each replicate and grown to 80% confluency before the addition of 4 μL 1 M 4SU to each plate. After a 14 h incubation, the media was aspirated; the cells were washed once with 5 ml ice-cold PBS for each plate and crosslinked by 0.15 J cm⁻² 365 nm UV light twice when on ice. The crosslinked cells were collected with cell lifters. 3 volumes of the lysis buffer (50 mM HEPES, pH 7.5; 150 mM KCl; 2 mM EDTA; 0.5% (v/v) NP-40, with 1:100 protease inhibitor (Roche) and 40 U ml⁻¹ RNasin® Ribonuclease Inhibitors (Promega) added freshly) was added to the cell pellet and incubated on ice for 10 min with periodic perturbation. The cell lysate was then centrifuged at 15,000 g for 15 min and the clear supernatant was collected. RNase T1 (1000 U μL⁻¹, Thermo Fisher Scientific) was added to the clear lysate to a final concentration of 0.1 U μL⁻¹ and an incubation at room temperature was performed for 15 min. The reaction was then quenched on ice. After 5 min, antibodies were added to each sample (5 μg or according to the manufacturer's protocols). The antibody and the lysate were incubated at 4 °C for 2 h under periodic rotation. Protein A beads or protein G beads (Thermo Fisher Scientific) were washed (50 μL for each sample, or adjusted to the antibodies amount) with IP

wash buffer (50 mM HEPES, pH 7.5, 300 mM KCl, 0.05% (v/v) NP-40, with 1:100 protease inhibitor (Roche) and 40 U ml$^{-1}$ RNasin® Ribonuclease Inhibitors (Promega) added freshly) for 2 times. The beads were resuspended in 50 μL lysis buffer for each sample and added to the antibody-lysate mixture subsequently. Another 2-h incubation at 4 °C was performed with low-speed rotation.

After the incubation, the beads were washed three times with IP wash buffer (50 mM HEPES, pH 7.5, 300 mM KCl, 0.05% (v/v) NP-40, with 1:100 protease inhibitor (Roche) added freshly) and then resuspended with 200 μL IP wash buffer per sample. The beads were treated with a second round of RNase T1 digestion under a final concentration of 10 U μL$^{-1}$ for 15 min at room temperature. The reaction was then quenched with the addition of 10 μL SUPERase-In followed by a 5-min incubation on ice. The beads were washed three times with high-salt wash buffer (50 mM HEPES, pH 7.5, 500 mM KCl, 0.05% (v/v) NP-40, with 1:100 protease inhibitor (Roche) added freshly) and twice with 1X PNK buffer (NEB) afterward. The beads were resuspended with 200 μL of 1X PNK buffer (NEB) and underwent T4 PNK (Thermo Fisher Scientific) end repair with standard procedures as previously mentioned under 37 °C. After the incubation, the beads were washed once with 1X PNK buffer followed by proteinase K digestion as described before. The RNA was recovered with RNA Clean & Concentrator (Zymo Research) before library construction by NEBNext Small RNA Library Prep Set for Illumina (NEB). All libraries were sequenced on Illumina Nova Seq with single-end 100 bp read length.

### ART-seq (active mRNA translation sequencing)
Two days after transient knockdown of *IGF2BP3* in HepG2 cell in six-well plate, cycloheximide was added with the final concentration of 100 μg/mL. The HepG2 cell was incubated at 37 °C for 4 min. The cells were washed twice with 0.5 mL DPBS with 100 μg/mL cycloheximide before added 0.4 mL lysis buffer (20 mM Tris-HCl pH 7.5, 100 mM KCl, 5 mM MgCl2, 0.5% NP-40, 1 mM DTT, 0.1 mg/mL cycloheximide, 0.1 U/μL Invitrogen Turbo DNase). The cells were scraped and collected into 1.5 mL tubes, and then incubated on ice with periodic inversions for 10 min. The lysate was clarified by centrifugation for 10 min at 20,000 x g at 4 °C. 0.3 mL supernatant was transferred to a fresh tube. 300 U of RNAse I was added, and the lysate was incubated at room temperature for 45 min. Afterwards, 15 μL Invitrogen SUPERase·In™ RNase Inhibitor (20 U/μL) to stop this reaction. The 0.3 mL of lysate was transferred to a prewashed MicroSpin S-400 column (Sigma: Microspin™ S-400 HR). The column was centrifuged at 600 x g for 2 min to collect flow-through. Zymo RNA Clean & Concentrator Kit was used to purify RNA. The 28–30 nt RPFs (ribosome-protected fragments) were excised from 10% TBU gel and recovered by ZR small-RNA™ PAGE Recovery Kit. The rRNA depletion was performed and the RPFs were end-repaired to have a 5′-phosphate and 3′-OH by T4 PNK treatment. The library was prepared with NEB small RNA library preparation kit.

### RNA-seq analysis
Raw reads were trimmed with Trimmomatic-0.39[68], then aligned to human genome and transcriptome (hg38) using HISAT (version 2.1.0)[69] with '-rna-strandness RF' parameters. Annotation files (version v29, 2018-08-30, in gtf format) were downloaded from GENCODE database (https://www.gencodegenes.org/).

### MeRIP-seq analysis
Mapped reads were separated by strands with samtools (version 1.9)[70] and m⁶A peaks on each strand were called using MACS (version 2)[71] with parameter '-nomodel, –keep-dup 5, -g 1.3e8 and -extsize 150' for m⁶A MeRIP-seq, and '-nomodel, –keep-dup 5, -g 1.3e8 and -extsize 75' for m⁷G MeRIP-seq separately. Significant peaks with $q < 0.01$ identified by MACS2 were considered. Peaks identified in at least three biological

replicates were merged using bedtools (v.2.26.0)[70] and were used in the following analysis.

### Half lifetime total RNA-seq analysis
Mapped reads on each GENCODE annotated gene were counted using HTSeq[72] and then normalized to counts per million (CPM) using edgeR[73] packages in R. CPM was converted to attomole by linear fitting of the RNA ERCC spike-in. Half lifetime of RNA was estimated using formula listed in previously published paper[4]. Specifically, as actinomycin D treatment results in transcription stalling, the change of RNA concentration at a given time (dC/dt) is proportional to the constant of RNA decay ($K_{decay}$) and the RNA concentration (C), leading to the following equation:

$$\frac{dC}{dt} = -K_{decay}C \tag{1}$$

Thus, the RNA degradation rate $K_{decay}$ was estimated by:

$$\ln\left(\frac{C}{C_0}\right) = -K_{decay}t \tag{2}$$

To calculate the RNA half-life ($t_{1/2}$), when 50% of the RNA is decayed (that is, $\frac{C}{C_0} = \frac{1}{2}$), the equation was:

$$\ln\left(\frac{1}{2}\right) = -K_{decay}t_{\frac{1}{2}} \tag{3}$$

From where:

$$t_{\frac{1}{2}} = \frac{ln2}{K_{decay}} \tag{4}$$

The final half-life was calculated by using the average values of 0 h, 3 h and 6 h.

### ART-seq analysis
Raw reads were trimmed using Cutadapt[74] to remove low-quality bases and adapters. The trimmed reads were then aligned to the human genome (hg38) using STAR[75]. Reads for each gene were counted using featureCounts[76]. Translation efficiency (TE) was calculated as the ratio of ribosome-protected fragments (RPFs) to RNA abundance, as determined by ART-seq and RNA-seq and identified using DESeq2[77].

### PAR-CLIP analysis
Low quality reads were filtered using 'fastq_quality_filter', and adapter were clipped using 'fastx_clipper', then adapter-free were collapsed to remove PCR duplicates by using 'fastx_collapser' and finally reads longer than 15 nt were retained for further analysis (http://hannonlab.cshl.edu/fastx_toolkit/). Preprocessed reads were mapped using bowtie with '-v 3 -m 10 –best –strata' parameters. Peaks were called using PARalyzer[78] software.

### Reporting summary
Further information on research design is available in the Nature Portfolio Reporting Summary linked to this article.

## Data availability
The data supporting the findings of this study are available from the corresponding authors upon request. m⁷G-MeRIP-seq in WT HepG2 cells, knockdown control cells, and METTL1 stable knockdown HepG2 cells, and m⁷G-seq in HepG2 cells were downloaded and re-analyzed in this study (GSE112276). The sequencing data produced in this study have been deposited in Gene Expression Ominibus (GEO) repository under the accession number GSE241222. The mass spectrometry

proteomics data generated in this study have been deposited to the ProteomeXchange Consortium via the PRIDE[79] partner repository with the dataset identifier PXD049390 and 10.6019/ PXD049390. The list of top proteins enriched by m7G probes based on the proteomics data are provided in Supplementary Data 1. The primers for dCas13b construction, guide RNA target sequences, and qPCR primers used in this study are summarized in Supplementary Data 2. Source data are provided with this paper.

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

## Acknowledgements

This work was supported by R01 HL155909 (C.H.) and RM1 HG008935 (C.H.). The Mass Spectrometry Facility of the University of Chicago is funded by the National Science Foundation (CHE-1048528). C.H. is an investigator of the Howard Hughes Medical Institute. We thank Dr. Jianjun Chen (City of Hope) for gifting plasmids for IGF2BP family proteins with KH domain and RRM domain modified. We thank Dr. Pieter W. Faber and Genomics Facility of the University of Chicago for their generous help with high-throughput sequencing. This article is subject to HHMI's Open Access to Publications policy. HHMI lab heads have previously granted a nonexclusive CC BY 4.0 license to the public and a sublicensable license to HHMI in their research articles. Pursuant to those licenses, the author-accepted manuscript of this article can be made freely available under a CC BY 4.0 license immediately upon publication.

## Author contributions

C.L. and C.H. conceived the project. C.L. and C.H. planned the experiments. C.L. designed and executed most experiments, analyzed the

data, and produced figures. X.D. analyzed high throughput data and performed bioinformatic analyses. Y.Z. executed experiments, analyzed the data, and produced figures. L.Z., L.-S.Z., Q.D., J. L., T.W., and Y.X. contributed to experiments including cell proliferation, cell cycle, probe synthesis and data analysis. C.L. and C.H. wrote the manuscript. All authors contributed to the discussion of the results.

## Competing interests

C.H. is a scientific founder, a member of the scientific advisory board and equity holder of Aferna Bio, Inc. and Ellis Bio, Inc., a scientific cofounder and equity holder of Accent Therapeutics, Inc., and a member of the scientific advisory board of Rona Therapeutics and Element Biosciences. The remaining authors declare no competing interests.
