## [Peer Review File · Nature Communications]

IGF2BP3 promotes mRNA degradation through internal m7G modificationREVIEWER COMMENTS

Reviewer #1 (Remarks to the Author):

Liu et al. reported IGF2BP proteins, which were previously identified as m6A reader proteins, are also capable of recognizing and binding to another type of internal mRNA methylation, namely m7G, and possess distinct functions in cancer cells based on their different preferences to m6A and m7G. Specifically, IGF2BP3 showed greater responsiveness to m7G modification and facilitated the degradation of m7G transcripts, whereas IGF2BP1 prefers m6A and stabilizes the m6A transcripts. Mechanistically, IGF2BP3 induces mRNA decay by interacting with EXOSC2 and XRN2. In the case of glioma, the interaction between IGF2BP3 and m7G plays a significant role in regulating the stability of TP53, which in turn affects tumor cell growth and chemoresistance. Overall, the discoveries presented in this manuscript are noteworthy and aligned with the scope of Nature Communications. However, I have several queries that must be addressed by the authors prior to manuscript acceptance.

1. Figures 3a and 3b demonstrate that the knockdown of IGF2BP3 caused a significant yet subtle decrease in the stability of m7G transcripts on a global level. This raises doubts about IGF2BP3's role as a predominant inducer of mRNA decay. Given that METTL1-mediated m7G modification was implicated in promoting translation, it is worthwhile to investigate the impact of IGF2BP3 on the translation of its target genes via m7G. Furthermore, it is crucial to scrutinize the expression of IGF2BP3-m7G target genes, including TP53, NDRG1, and ZDHHC9, both at the mRNA and protein levels in cells with either overexpression or knockdown of IGF2BP3. It is particularly important to validate the expression of TP53 protein, as proteins are the primary mediators of cellular function.
2. What is the mechanism that mediates the different preferences of IGF2BP proteins towards m6A and m7G?
3. This manuscript demonstrates that IGF2BP1 and IGF2BP3 both interact with the exosome complex and mediate mRNA decay of m7G transcripts. However, it is still unclear why m6A transcripts have an opposite fate, which is supposed to be associated with IGF2BPs-exosome complex as well.
4. Does overexpression of IGF2BP3 in glioma show similar effects on tumor growth and chemoresistance as *dcas-IGF2BP3+sgTP53*?
5. LN229 cells harbor a mutated form of TP53, specifically P98L, which does not affect the DNA-binding ability of TP53 but may have an unknown effect on its function. Therefore, it would be advisable to include at least one glioma cell line that expresses wildtype TP53 to validate the function of IGF2BP3.
6. All of the co-IP experiments should include proper negative controls.
7. The statement "T98G with mutant p53 but gain of function" on page 6, line 113 is inaccurate. T98G cells express a M237I mutant form of TP53, which leads to decreased DNA binding and TP53 transactivation activity and is considered a "loss of function" mutation (<https://ckb.jax.org/geneVariant/show?geneVariantId=16637>, PMID: 16492679, PMID: 20080630, PMID: 31395785). Please make the necessary correction.

Reviewer #2 (Remarks to the Author):

This manuscript by Liu et al suggest that IGF2BP RNA-binding proteins are cellular readers of internal m7G modification in mRNA. They rightly point to IGF2BP3 being the family member with the greatest regulatory potential on m7G modified transcripts. Additionally, they point to a role for IGF2BP3 and the m7G-writer METTL1 in regulating TP53 mRNA, leading to modulation of cancer progression and chemosensitivity.

Major Points:

1) For most pieces of data presented the number of independent replicates performed seems to be only 2. This is quite unacceptable – three is the minimal number of experiments required, especially for key data presented in main figures. Additionally, there seem to be some discrepancies in reporting – for example Fig. 5d shows three data points, but the legend indicates 2. These discrepancies need to be clarified. Similarly, the number of replicates performed for sequencing-based experiments (IGF2BP KD for half-life), and PARCLIP are unclear as this reviewer does not have a secure token for the GSE deposition. This is important, especially for PAR-CLIP as fewer replicates is known to lead to an overrepresentation of false positives.

2) The proteomic data (from Fig. 2a) much also be deposited in public proteomic repositories as this is an NIH-funded study.

3) Right now, the authors have shown that IGF2BP3 and METTL1 (m7G) regulate the same transcripts. However, the conclusive proof linking IGF2BP3 and m7G is missing. Given the beautiful dCas13 targeting system that the authors have convincingly shown works, the authors have a unique opportunity to improve on this. This manuscript (and the link between IGF2BP3 and m7G) would be greatly strengthened by testing what happens when dCas13 targeting of IGF2BP3 or METTL1 are coupled with depletion of the other factor. The following three experiments are essential in my opinion:

- a. Targeting dCas13-IGF2BP3 to a transcript upon siCTRL vs siMETTL1 depletion and measuring RNA stability.
- b. Targeting dCas13-METTL1 to a transcript upon siCTRL vs siIGF2BP3 depletion and measuring RNA stability.
- c. Targeting dCas13-METTL1 to a transcript and measuring IGF2BP3 binding by RIP/CLIP-RT-qPCR.

Minor points:

1) The differences between G and m7G in the EMSA data (extended data 2b) is very very weak. However, coupled with the probe pulldown assays, and the rest of the manuscript, I do not view this as a major problem

2) The figure legends do not do a good job of explaining experimental specifics and should be revised so that readers can get a clear understanding. Alternatively, more detail needs to be included in the results section.

Reviewer #3 (Remarks to the Author):

Liu et al. have identified IGF2BPs as RNA-binding proteins that recognize m7G. Their study also demonstrates a specific interaction between IGF2BPs and m7G, which leads to the rapid degradation of target mRNAs. In addition, the authors have highlighted the functionally significant role of IGF2BPs in m7G-modified transcripts in glioblastoma cells. The manuscript contains a wealth of informative and intriguing data. However, there are two major concerns that have been raised (particularly see comments #15 and #16). Therefore, I recommend that the following comments are addressed thoroughly.

1. Figure 1a,b: What are the criteria for low expression and high expression of METTL1 in various cancers? This reviewer is also curious if the difference in survival probability between low and high METTL1-expressing cancers is statistically significant.
2. Figure 1e,f: The gene image for TP53 depicted in blue at the bottom lacks orientation information. In addition, information on the y-axis is missing (TPM, FPKM, etc). In Figure 1f, although shMETTL1 reduced the number of m7G IP reads mapped to the 3'UTR of TP53 mRNA, it looks like the effect of METTL1 knockdown is very marginal.
3. Figure 1g: The order of columns should be changed. It is now presented in the opposite order to the column description. The first column should be m6A, and the second column should be m7G for easy understanding.
4. Pages 112-114: Please describe the properties of all cell lines in more detail. The description, such as "mutant but functional," is not sufficient.
5. Extended Data Fig. 1e: It would be better if western blots showing endogenous p53 protein are included.
6. Figure 2a: Please provide information on the complete list of proteins obtained from mass spectrometry. What are the top-ranked proteins?
7. Extended Data Fig. 2b: What are the Kd values of each protein?
8. Extended Data Fig. 2d: Provide quantitative data showing the relative binding efficiencies of each variant. What is KH3-4 μ ? Is this the wild-type (full-length) protein lacking KH3-4 or the full-length protein containing point mutations in the KH3-4 domain?
9. Page 175: "Consistent with the previous report," cite a proper reference.
10. Page 180: "with most overlapping sites at the 3' UTR compared to the non-overlapped ones (Fig. 2e, left in blue)." Which line indicates non-overlapped ones in Figure 2e?
11. Extended Data Fig. 4b: Please insert a box plot as an inset, as presented in Figure 3a.
12. Extended Data Fig. 4c: For simple comparison, it would be better if the analysis of IGF2BP2 KD data is included in this figure. In addition, how did the authors calculate "IGF2BPs binding intensities" on page 209?
13. Page 210: Please provide IGV plots for the representative transcripts.
14. Figure 3e: Statistical calculations with p-value or r-value are missing.
15. Figure 3: In this figure, the authors characterized and tried to validate different roles of IGF2BPs in m6A and m7G-modified mRNAs. Although the present data are supportive of the claim, it is still not conclusive. To completely rule out the m6A effect on mRNA stability, the reviewer strongly recommends that all (or at least some) experiments should be done under conditions lacking the m6A writer. Otherwise, a possible interplay between m6A and m7G cannot be excluded, although the authors showed an insignificant correlation of

methylation levels between m7G and m6A at the transcript level (Fig. 3e) and the limited overlapping peaks between both modifications in the 3'UTR (Fig. 3f).

16. Figure 4: When IGF2BPs are tethered to the mRNAs, one predicts two possible consequences: First, based on IGF2BPs-mediated m6A mRNA stabilization, the tethering of IGF2BPs may stabilize the mRNAs. Alternatively, considering that IGF2BPs trigger rapid degradation of m7G-modified mRNAs, their tethering may destabilize the mRNAs. Although the authors targeted endogenous mRNAs containing m7G residue, some sequences in proximity to the m7G residue may be m6A-modified. Therefore, as mentioned in comment #15, this reviewer strongly recommends that the tethering experiments should be conducted under conditions depleted of the m6A writer.

17. Figure 4f: A negative control lane is missing.

18. Figure 4g: The data presented in this figure do not support the authors' claim. Statistical analysis is necessary.

19. Figure 5d,e: This reviewer is curious why overexpression or knockdown of IGFBP3 affects the m7G level of p53 mRNA? Does this protein directly affect m7G modification?"

Summary

We sincerely thank all the reviewers for their constructive comments, which have significantly improved our manuscript. Specifically, all reviewers suggested experiments to further validate our regulatory model, which has strongly strengthened our conclusion and the biological significance.

Reviewer #2 listed a series of experiments taking advantage of our dCas13b tethering system to validate the coupling effect of IGF2BP3 and m⁷G, regarding the m⁷G-dependent regulation of mRNA degradation by IGF2BP3. In the control cells, we consistently observed accelerated degradation upon tethering with either IGF2BP3 or METTL1. However, knocking down either *IGF2BP3* or *METTL1* attenuated the effects from the introduction of the dCas13b tethering system. Moreover, when we tethered our representative targets with dCas13b-METTL1, we noticed a significant increase in IGF2BP3 binding. These findings underscore that both IGF2BP3 and RNA methylation are critical to mRNA half-life regulation through m⁷G.

Reviewer #3 highlighted the importance of validating our model in the context with the depletion of METTL3, m⁶A writer, to further evaluate potential impacts of m⁶A on m⁷G targets through IGF2BP3. We accordingly generated a cell line with stable downregulation of *METTL3* and performed the decay assay in sh*METTL3* HepG2 with transient *IGF2BP1-3* knockdown, respectively. We observed a stabilization effect on m⁷G targets upon *IGF2BP1* and *IGF2BP3* knockdown, consistent with our previous results in wild-type cells. We further applied the dCas13b tethering system in the sh*METTL3* cells and found that the knockdown of *METTL3* didn't interfere with the degradation promotion of dCas13b-METTL1 or IGF2BP3 on our representative targets. These data further supported our conclusion that IGF2BP3 could promote degradation of m⁷G-targets independent of m⁶A methylation.

Reviewer #1 suggested the inclusion of an additional cell line with wild-type TP53 for functional studies. To address this, we took advantage of the U87MG cell line already included in our study and conducted cell cycle and proliferation experiments. To our delight, results obtained from U87MG were consistent with those observed in LN229, and thus reinforcing our conclusion on the roles of m⁷G modification on *TP53* transcripts in cancer.

Collectively, the incorporation of diverse perspectives from all three reviewers has significantly strengthened our model that IGF2BP3 promotes the degradation of m⁷G-targets in the context of cancer regulation.

Detailed responses to address every comment from each reviewer are shown below.

Reviewer #1 (Remarks to the Author):

Liu et al. reported IGF2BP proteins, which were previously identified as m⁶A reader proteins, are also capable of recognizing and binding to another type of internal mRNA methylation, namely m⁷G, and possess distinct functions in cancer cells based on their different preferences to m⁶A and m⁷G. Specifically, IGF2BP3 showed greater responsiveness to m⁷G modification and facilitated the degradation of m⁷G transcripts, whereas IGF2BP1 prefers m⁶A and stabilizes the m⁶A transcripts. Mechanistically, IGF2BP3 induces mRNA decay by interacting with EXOSC2 and XRN2. In the case of glioma, the interaction between IGF2BP3 and m⁷G plays a significant role in regulating the stability of TP53, which in turn affects tumor cell growth and chemoresistance. Overall, the discoveries presented in this manuscript are noteworthy and aligned with the scope of Nature Communications. However, I have several queries that must be addressed by the authors prior to manuscript acceptance.

1. Figures 3a and 3b demonstrate that the knockdown of IGF2BP3 caused a significant yet subtle decrease in the stability of m⁷G transcripts on a global level. This raises doubts about IGF2BP3's role as a predominant inducer of mRNA decay. Given that METTL1-mediated m⁷G modification was implicated in promoting translation, it is worthwhile to investigate the impact of IGF2BP3 on the translation of its target genes via m⁷G. Furthermore, it is crucial to scrutinize the expression of IGF2BP3-m⁷G target genes, including TP53, NDRG1, and ZDHHC9, both at the mRNA and protein levels in cells with either overexpression or knockdown of IGF2BP3. It is particularly important to validate the expression of TP53 protein, as proteins are the primary mediators of cellular function.

Response: We thank the reviewer for the very positive comments.

We conducted active mRNA translation sequencing (ART-seq) to capture actively translated transcripts upon *IGF2BP3* knockdown compared to control. As expected, the knockdown of *IGF2BP3* resulted in decreased translation efficiency of transcripts with m⁶A modifications (Fig. 1a, green and orange groups compared to the control group in grey), suggesting IGF2BP3 could promote translation of m⁶A modified transcripts, consistent with previous reports¹. However, we observed a slight increase in the translation efficiency of m⁷G targets upon *IGF2BP3* knockdown. While m⁷G has been shown to promote translation², our results seem to suggest complicated but mild effects.

Furthermore, we quantified mRNA and protein expression levels of related targets using RT-qPCR and western blotting in various cell lines as the reviewer pointed out. In HepG2, the knockdown of *IGF2BP3* not only stabilized all transcripts as described but also led to increased mRNA levels of HUWE1, CYRIB, and, notably, TP53, with the remaining targets showing little changes (Fig. 1b). We also observed an increase in the protein expression levels of HUWE1 and TP53 (Fig. 1c).

We expanded our analysis to other cancer cell lines. We observed a significant increase in TP53 levels upon *IGF2BP3* knockdown in cells with TP53 highly m⁷G-methylated at its 3'UTR (A172 and T98G) (Fig. 1d). Consistently, in cancer cells where TP53 is lowly methylated (LN229 and U87MG), the overexpression of IGF2BP3 induced downregulation of TP53 transcripts levels and protein levels (Fig. 1e).

These findings collectively indicate that IGF2BP3 promotes the degradation of m⁷G targets, potentially affecting RNA and protein levels, particularly for TP53. We think m⁷G may also impact translation regulation, and the readers and pathways associated with these processes could be further investigated. We have summarized these results in Fig. 1a-e as data for the reviewers' reference below and intend to explore these aspects in the future research.

Figure 1 for Review 1: IGF2BP3 promotes the degradation of m⁷G targets and affects protein levels. (a) Cumulative distribution \log_2FC (fold changes) of translation efficiency (ratio of ribosome bound fragments to mRNA input) between si*IGF2BP3* and siControl transfection in HepG2 cells, with groups of all gene (gray), targets with only m⁷G modification (blue), targets with only m⁶A modification (orange), and targets with both modifications (green). **(b)** RT-qPCR results of relative mRNA expression level of the representative m⁷G-modified targets upon *IGF2BP3* knockdown in HepG2 cells. Mean \pm SEM of three independent experiments. Two-tailed Student's t-tests were used.

P values are marked at the top of each group. (c) Western blotting results of relative protein expression level of the representative m⁷G-modified targets upon *IGF2BP3* knockdown in HepG2 cells. (d) RT-qPCR results of relative TP53 mRNA expression level (left) and western blotting results of relative TP53 protein expression level (right) upon *IGF2BP3* knockdown in the cell lines with highly methylated TP53 transcripts. Mean ± SEM of three independent experiments for RT-qPCR. Two-tailed Student's t-tests were used. *P* values are marked at the top of each group. (e) RT-qPCR results of relative TP53 mRNA expression level (left) and western blotting results of relative TP53 protein expression level (right) upon *IGF2BP3* overexpression in the cell lines with lowly methylated TP53 transcripts. Mean ± SEM of three independent experiments for RT-qPCR. Two-tailed Student's t-tests were used. *P* values are marked at the top of each group. Western blotting results also confirmed the overexpression of *IGF2BP3* with Flag tag.

2. What is the mechanism that mediates the different preferences of IGF2BP proteins towards m⁶A and m⁷G?

Response: We thank the reviewer for highlighting this important point. While IGF2BP proteins share high similarity in domain arrangement, they exhibit differences in structure and motif preferences. Taking advantage of available structural studies on the IGF2BP proteins and our MeRIP-seq data, we calculated the possible enrichment of m⁶A and m⁷G modifications across motifs preferred by the three proteins (Fig. 2a). We observed significant differences among the family members; m⁷G transcripts are preferred by IGF2BP3 while m⁶A are more bound by IGF2BP1 and IGF2BP3, suggesting their distinct binding preferences for transcripts with different modifications.

We also performed *in vitro* binding experiments by overexpressing IGF2BP family proteins with mutations of GxxG to GEEG in the KH domains to disrupt the binding domain. As the western blotting showed (Fig. 2b, left), along with the relative quantification results compared to the wild-type conditions (Fig. 2b, right), mutations in KH1-4 abolished the m⁷G binding (first column in Fig.2b left, not observable in Fig.2b right) while deletion of the RRM domain didn't change the binding preference on m⁷G much (fourth column in Fig.2b left, the green column in Fig. 2b right), suggesting that these KH1-4 regions are responsible for m⁷G binding. While KH1-2 mutations also retained the binding preference, mutated KH3-4 in IGF2BP1 and IGF2BP2 entirely disrupted their binding on m⁷G probes. However, we noticed that mutated KH3-4 in IGF2BP3 showed partial binding preference on m⁷G probes (bottom row of the third column in Fig. 2b left), implying that KH1-2 domains in IGF2BP3 might help its binding to m⁷G. Interestingly, previous reports showed that all three family members with mutated KH3-4 domains could not bind to m⁶A probes, indicating that at least for IGF2BP3, it binds m⁶A and m⁷G in a slightly different way, given the difference in the involvement of the KH1-2 domains.

The binding affinities of the three proteins may also be affected by local structural features near RNA modifications. Structural studies of the proteins binding to specific targets might be required. Given that our current study primarily focuses on the function roles of mRNA internal m⁷G modification, we did not delve into additional details. We have accordingly updated the Discussion Part (line 440-443) to reflect this point raised by the reviewer.

Figure 2 for Review 1: Interactions of IGF2BPs with m⁷G and m⁶A. (a) Enrichment of m⁶A (left) or m⁷G (right) with IGF2BP1, IGF2BP2 and IGF2BP3 motifs, respectively, quantified with Odds Ratio. (b) Methylated RNA probe pull-down followed by western blotting showed *in vitro* binding of the baits with KH domain-mutated IGF2BP variants and RRM domain-depleted IGF2BP2. Left: western blotting showing the enrichment of m⁷G probes with each variant. Right: quantification of relative enrichment of m⁷G probes compared to those in wild-type cell lysates. KH1-2, KH3-4, and KH1-4, refer to mutation of GxxG to GEEG in corresponding KH domains in the full-length proteins. (c) Left: metagene plot of IGF2BPs binding sites (not overlapped with m⁷G modified sites) (orange), m⁷G modified sites (not overlapped with IGF2BPs binding sites) (blue), and their overlapped sites (green). Right: metagene plot of IGF2BPs binding sites (not overlapped with m⁶A modified sites) (orange), m⁶A modified sites (not overlapped with IGF2BPs binding sites) (light green), and their overlapped sites (purple).

3. This manuscript demonstrates that IGF2BP1 and IGF2BP3 both interact with the exosome complex and mediate mRNA decay of m⁷G transcripts. However, it is still unclear why m⁶A transcripts have an opposite fate, which is supposed to be associated with IGF2BPs-exosomes complex as well.

Response: While we do not yet possess a definitive answer, we have formulated several hypotheses to address this intriguing point. Firstly, both m⁶A and m⁷G modifications recruit a large variety of reader proteins that may "compete" for binding to their respective targets, guided by factors such as binding affinity and interaction intensity. For instance, m⁶A-modified transcripts may be susceptible to degradation and translation regulation by the YTH family proteins^{3, 4}, whereas m⁷G-modified transcripts could be shuttled by QKI proteins to facilitate regulation under stress conditions⁵. The dynamics of transcripts bearing these modifications may vary, even when they are predisposed to bind the same protein, due to changes in the abundance or activity of these regulatory factors.

Moreover, considering m⁶A modification, existing studies have delineated its involvement in both stabilization (e.g., via IGF2BP proteins¹) and destabilization (e.g., via YTHDF2³). Alterations in local motifs and structural arrangements near a modified site may readily modulate its access to specific reader proteins. Additionally, the localization of binding sites, whether within

exons or untranslated regions (UTRs), could significantly influence the regulatory pathways engaged. Interestingly, our analysis revealed that while the interactions between IGF2BPs and m⁶A or m⁷G predominantly occur in the 3'UTR, the overlapped peaks of IGF2BPs and m⁶A are mainly concentrated near the stop codon (Fig. 2c, right, purple), indicating a more focused distribution. Conversely, interactions with m⁷G span across the entire 3'UTR (Fig. 2c, left, green), implying that IGF2BP proteins might bind to different modifications at distinct regions within the 3'UTR, thereby recruiting varied partners for RNA regulation.

The interplay between readers and RNA modifications is highly context dependent. For instance, YTHDC1, a nuclear m⁶A reader, facilitates exon inclusion of its targets by recruiting pre-mRNA splicing factor SRSF3⁶. It also mediates the export of methylated transcripts from nucleus to cytoplasm⁷. Recent studies have implicated YTHDC1 in the degradation of chromatin-associated RNA through its interaction with the Nuclear Exosome Targeting (NEXT) complex⁸. The diverse regulatory roles of YTHDC1 are contingent upon the specific RNA targets and protein partners it interacts with. Additionally, modification sites and associated regulatory pathways exhibit cell-type specificity. While we observed stabilization of m⁶A by IGF2BP proteins in the HepG2 cells, consistent with previous findings in HEK293T cells, these dynamics may vary in other cell types. We thank the reviewer for the insightful question and have updated the Discussion Part (line 450-462) accordingly.

4. Does overexpression of IGF2BP3 in glioma show similar effects on tumor growth and chemoresistance as dCas-IGF2BP3+sgTP53?

Response: We evaluated the impact of IGF2BP3 on cell growth and chemoresistance. In both LN229 and U87MG cells, the overexpression of IGF2BP3 protein promoted cell proliferation (Fig 2, left and middle). IGF2BP3 overexpression also sensitized T98G cells to TMZ treatment, reducing their chemoresistance (Fig 2, right). Importantly, these observations are consistent with the outcomes obtained from the tethering system utilizing dCas13b-IGF2BP3, reinforcing the reliability of our regulatory model across different manipulation strategies. The figures have been included in revised Extended Data Fig. 9k and 9o.

Figure 3 for Reviewer 1: IGF2BP3 overexpression in glioma shows similar effects on tumor growth and chemoresistance as dCas13b-IGF2BP3 with sgTP53. Left and middle: cell proliferation in LN229 (left) and U87MG (middle) cells with IGF2BP3 overexpression. Mean \pm SEM of four independent experiments. Two-tailed

Student's t-tests were used. Right: survival rate of the T98G cells with IGF2BP3 overexpression treated with different concentrations of TMZ. Mean \pm SEM of four independent experiments. Two-tailed Student's t-tests were used.

5. LN229 cells harbor a mutated form of TP53, specifically P98L, which does not affect the DNA-binding ability of TP53 but may have an unknown effect on its function. Therefore, it would be advisable to include at least one glioma cell line that expresses wildtype TP53 to validate the function of IGF2BP3.

Response: This is a good point. Per the suggestion, we incorporated U87MG, a cell line used in our study with wild-type TP53, and performed the relevant function analyses, including its proliferation and its escape from G1 arrest upon the introduction of dCas13b-IGF2BP3 with the guide RNA for TP53. Encouragingly, all results obtained in U87MG are consistent with our findings in LN229, further supporting our proposed model, strengthening the robustness of our study. We have added the data to the revised Extended Data Fig. 9i, 9j, and 9m.

Figure 4 for Reviewer 1: U87MG presented a similar response in cell growth and cell cycle with introduction of dCas13b system on TP53 3'UTR. (a) Cell proliferation in U87MG cells with dCas13b-IGF2BP3 tethering with the guide RNA or the negative gRNA. Mean \pm SEM of six independent experiments, and two-tailed Student's t-tests were used. **(b)** Cell proliferation in U87MG cells with siTP53 compared to siControl. Mean \pm SEM of six independent experiments, and two-tailed Student's t-tests were used. **(c,d)** Distributions **(c)** and the normalized percentages **(d)** of the cells in the G0/G1, S, and G2/M phases in the cells with dCas13b-IGF2BP3 tethering.

6. All of the co-IP experiments should include proper negative controls.

Response: We have rerun all the co-IP experiments and included both rabbit IgG as negative control and IGF2BP2, the other IGF2BP family member, as another reference. All the blots (Fig. 4f, Extended Data Fig. 7a and 7b) have been updated accordingly in support of the original conclusions. We believe that these additions enhance the rigor and reliability of our experimental findings.

7. The statement "T98G with mutantTP53 but gain of function" on page 6, line 113 is inaccurate. T98G cells express a M237I mutant form of TP53, which leads to decreased DNA binding and Tp53 transactivation activity and is considered a "loss of function" mutation (<https://ckb.jax.org/geneVariant/show?geneVariantId=16637>, PMID: 16492679, PMID: 20080630, PMID: 31395785). Please make the necessary corrections.

Response: We apologize for the misleading description. We have revised our wording to provide more details for both T98G and LN229 cells at line 116-120.

Reviewer #2 (Remarks to the Author):

This manuscript by Liu et al suggests that IGF2BP RNA-binding proteins are cellular readers of internal m⁷G modification in mRNA. They rightly point to IGF2BP3 being the family member with the greatest regulatory potential on m⁷G modified transcripts. Additionally, they point to a role for IGF2BP3 and the m⁷G-writer METTL1 in regulating TP53 mRNA, leading to modulation of cancer progression and chemosensitivity.

Major Points:

1) For most pieces of data presented the number of independent replicates performed seems to be only 2. This is quite unacceptable – three is the minimal number of experiments required, especially for key data presented in main figures. Additionally, there seem to be some discrepancies in reporting – for example Fig. 5d shows three data points, but the legend indicates 2. These discrepancies need to be clarified. Similarly, the number of replicates performed for sequencing-based experiments (IGF2BP KD for half-life), and PARCLIP are unclear as this reviewer does not have a secure token for the GSE deposition. This is important, especially for PAR-CLIP as fewer replicates is known to lead to an overrepresentation of false positives.

Response: We thank the reviewer for emphasizing the importance of replicates and the consistency of the figures and the legend. We have updated all related data with at least three replicates as well as the corresponding figure legends.

With regards to the sequencing data, we have deposited the sequencing data to GSE241222 (<https://www.ncbi.nlm.nih.gov/geo/query/acc.cgi?acc=GSE241222>), which can be accessed with temporary token as uhmzeakgpjqrqd.

We appreciate the emphasis on data quality and the importance of minimizing false positives. As suggested, we performed two additional replicates for PAR-CLIP of IGF2BP3 in HepG2 cells, the key regulator in our study. We successfully acquired peaks of comparable numbers and observed an around 40% overlap of the peaks between the two batches (Fig. 1a). We therefore continued to evaluate the consistency of our new batch data. Still, more than half of the IGF2BP3 targets are modified by internal m⁷G (Fig. 1b). The metagene profiles confirmed the overlap of the m⁷G peaks with IGF2BP-bound sites (Fig. 1c, left in green), with most overlapping sites at the 3' UTR (Fig. 1c, right) compared to the non-overlapped but methylated ones (Fig. 1c, left in blue). When examining their binding overlaps at the 3' UTR we observed a consistent about 10% of the IGF2BP3 binding sites adjacent to m⁷G peaks (Fig. 1d). Furthermore, utilizing the second batch of PAR-CLIP data, we consistently identified IGF2BP3 binding signals near the m⁷G peaks at our representative targets with little interference from m⁶A methylation. We have incorporated the data into IGV plots of these targets (Fig. 1e). Additionally, we conducted correlation analyses for the sequencing data of the decay assay (Fig. 1f) and observed a high correlation between each pair of replicates, suggesting the robustness of our conclusions. We have incorporated these analyses into the revised Extended Data Fig. 4b and 4e.

Figure 1 for Reviewer 2: Analyses for PAR-CLIP of IGF2BP3 and half lifetime decay assay. (a) Venn diagram of overlaps of transcripts that were bound by IGF2BP3 in the original batch and those from the new repeat batch. **(b)** Venn diagram of overlaps of transcripts that were bound by IGF2BP3 in the new repeat batch and modified with m⁷G. **(c)** Left: metagene plot in the new repeat batch of IGF2BP3 binding sites (not overlapped with m⁷G modified sites) (orange), m⁷G modified sites (not overlapped with IGF2BP3 binding sites) (blue) and their overlapping sites (green). Right: Lollipop plot showing the overlaps between m⁷G and IGF2BP3 in the new repeat batch quantified using Jaccard Index. **(d)** Venn diagram of overlaps of peaks that were bound by IGF2BP3 and modified with m⁷G in the 3' UTR in the new repeat batch. **(e)** IGV plots demonstrating the representative genes with m⁷G modification at the loci, bound by IGF2BP proteins, but with little m⁶A modification nearby. y axis showing counts per ten million reads. **(f)** Correlation analyses for each pair of replicates of half lifetime decay assays. From left to right: knockdown control, *siIGF2BP1*, *siIGF2BP2*, and *siIGF2BP3*.

2) The proteomic data (from Fig. 2a) much also be deposited in public proteomic repositories as this is an NIH-funded study.

Response: As suggested, we have submitted our dataset to ProteomeXchange via the PRIDE database. The submission is currently available through reviewer account with username as reviewer_pxd049390@ebi.ac.uk and password as bhDxhEvn. Upon publication, the dataset will be made publicly available. Additionally, we have included information about the proteomic data access in both the Results section (lines 140-142) and the Methods section.

3) Right now, the authors have shown that IGF2BP3 and METTL1 (m7G) regulate the same transcripts. However, the conclusive proof linking IGF2BP3 and m7G is missing. Given the beautiful dCas13 targeting system that the authors have convincingly shown works, the authors have a unique opportunity to improve on this. This manuscript (and the link between IGF2BP3 and m7G) would be greatly strengthened by testing what happens when dCas13 targeting of IGF2BP3 or METTL1 are coupled with depletion of the other factor. The following three experiments are essential in my opinion:

- a. Targeting dCas13-IGF2BP3 to a transcript upon siCTRL vs siMETTL1 depletion and measuring RNA stability.
- b. Targeting dCas13-METTL1 to a transcript upon siCTRL vs siIGF2BP3 depletion and measuring RNA stability.
- c. Targeting dCas13-METTL1 to a transcript and measuring IGF2BP3 binding by RIP/CLIP-RT-qPCR.

Response: We appreciate these suggestions. In response, we conducted all three sets of experiments as illustrated in Figure 2.

In the knockdown control cells for both scenarios (a and b), we consistently observed accelerated degradation upon tethering with either IGF2BP3 (Fig. 2a) or METTL1 (Fig. 2b). Conversely, in cells with either *IGF2BP3* or *METTL1* knockdown, we observed limited effects following the introduction of the dCas13b tethering system. These findings collectively highlight the significance of both IGF2BP3 and METTL1 in regulating mRNA half-life. Moreover, when we tethered our representative targets with dCas13b-METTL1, we noted a significant increase in IGF2BP3 binding (Fig. 2c), providing additional support for the intricate interplay between m⁷G modification and IGF2BP3. We extend our gratitude to the reviewer for the experimental proposal, as these results provide additional evidence that both IGF2BP3 and m⁷G modifications are indispensable for the regulation of transcript half-life. The data have been updated to the revised Extended Data Fig. 6c, 6e, and 6f accordingly.

Figure 2 for Reviewer 2: Both IGF2BP3 and m⁷G modifications are indispensable for the promotion of mRNA degradation. (a) Changes in mRNA levels in HepG2 cells upon *METTL1* knockdown with introduction of the dCas13b-IGF2BP3 and guide RNA at the target loci or not (neg). Mean ± SEM of two independent experiments. Two-tailed student's t-tests were used. Calculated half lifetime values are marked in the corresponding colors. *P* values are marked next to the calculated half lifetime values. (b) Changes in mRNA levels in HepG2 cells upon *IGF2BP3* knockdown with introduction of the dCas13b-METTL1 and guide RNA at the target loci or not (neg). Mean ± SEM of two independent experiments. Two-tailed student's t-tests were used. Calculated half lifetime values are marked in the corresponding colors. *P* values are marked next to the calculated half lifetime values. (c) RT-qPCR presenting the relative IGF2BP3 binding on the target loci upon the tethering of dCas13b-METTL1 with guide RNA and negative control guide RNA. Signals are normalized to input. Mean ± SEM of three independent experiments. Two-tailed student's t-tests were used. *P* values are marked for each group.

Minor points:

1) The differences between G and m⁷G in the EMSA data (extended data 2b) is very very weak. However, coupled with the probe pulldown assays, and the rest of the manuscript, I do not view this as a major problem.

Response: We thank the reviewer for pointing out the need to highlight the differences of binding affinity. We have calculated the K_d values for each binding based on the EMSA data and marked accordingly. By doing so, it is more evident to potential readers that the IGF2BP family proteins show a preference to m⁷G modification compared to G probes.

2) The figure legends do not do a good job of explaining experimental specifics and should be revised so that readers can get a clear understanding. Alternatively, more detail needs to be included in the results section.

Response: We appreciate the valuable suggestions from the reviewer to enhance the clarity of our experiments and related results. To facilitate better understanding, we have included additional details in both the figure legends and the Results sections of the revised manuscript.

Reviewer #3 (Remarks to the Author):

Liu et al. have identified IGF2BPs as RNA-binding proteins that recognize m7G. Their study also demonstrates a specific interaction between IGF2BPs and m7G, which leads to the rapid degradation of target mRNAs. In addition, the authors have highlighted the functionally significant role of IGF2BPs in m7G-modified transcripts in glioblastoma cells. The manuscript contains a wealth of informative and intriguing data. However, there are two major concerns that have been raised (particularly see comments #15 and #16). Therefore, I recommend that the following comments are addressed thoroughly.

1. Figure 1a,b: What are the criteria for low expression and high expression of METTL1 in various cancers? This reviewer is also curious if the difference in survival probability between low and high METTL1-expressing cancers is statistically significant.

Response: We want to thank the reviewer for careful reading of our manuscript. To clarify METTL1 expression level, we defined low and high expression of METTL1 in each cancer based on the median value. Specifically, those values lower than the median were categorized into the "Low" group, while those exceeding the median were grouped into the "High" category. To avoid any potential confusion, we have incorporated a detailed explanation of our criteria for METTL1 expression in the figure legend for Fig. 1b as “The ones lower than the median value are all grouped into the “Low” group while the ones higher than the median value are all grouped into “High”.

Figure 1 for Reviewer 3: Overall survival rate of representative cell lines with different expression levels of *METTL1* in tumor compared to normal tissue. From left to right: KICH, LAML, and THYM.

To comprehensively assess the impact of METTL1 expression on cancer survival rates, we analyzed various cancer types and presented three examples (Fig. 1): Thymoma (THYM), which exhibited elevated *METTL1* expression in tumors compared to normal tissues, and Kidney Chromophobe (KICH) and Acute Myeloid Leukemia (LAML) both showing similar levels of *METTL1* in tumor and normal tissues. Interestingly, we observed a significant correlation between high *METTL1* expression and low survival rate specifically in Leukemia, suggesting that the mRNA expression level of METTL1 may not be the sole determinant of this correlation.

In addition to mRNA levels, recent studies have evaluated *METTL1* gene activity as an indirect indicator of METTL1 protein expression⁹. The investigations have revealed inconsistency between mRNA and protein expression levels of METTL1 in certain cancers, which can be attributed to protein metabolism and post transcriptional modification. Correspondingly, although we did not observe differential expression of *METTL1* mRNA in LAML between tumor and normal tissues, other studies have reported upregulation of METTL1 protein levels in patients¹⁰.

In support of our findings, immunohistochemical staining (IHC) data from 80 tissue samples in other studies demonstrate high expression levels of both METTL1 and WDR4 in primary glioblastoma (GBM) tissue compared to normal cerebral tissue¹⁰, underscoring the importance of METTL1 and the related m⁷G methylation pathway in glioblastoma. We have included these results (line 91-92) to help support our point. We suppose that METTL1 protein levels might be a more reliable reference. However, a more comprehensive pan-cancer analysis would validate this hypothesis, which might be beyond the scope of our current study.

2. Figure 1e,f: The gene image for TP53 depicted in blue at the bottom lacks orientation information. In addition, information on the y-axis is missing (TPM, FPKM, etc). In Figure 1f, although shMETTL1 reduced the number of m⁷G IP reads mapped to the 3'UTR of TP53 mRNA, it looks like the effect of METTL1 knockdown is very marginal.

Response: We thank the reviewer's feedback on the unclear labeling in our figures. To address this, we have added orientation information for TP53 in Figures 1e-f and included the y-axis information in the figure legends.

We acknowledge the reviewer's feedback regarding the clarity of the IGV plots in presenting the impact of METTL1 reduction on TP53 3'UTR methylation. We also recognize the potential limitation that the generation of a stable knockdown cell line might attenuate the effect of METTL1 downregulation on TP53 methylation, given the pivotal role of TP53 in cellular regulation. To address these concerns, we performed m⁷G-MeRIP-qPCR with transient *METTL1* knockdown in HepG2 cells and noted a significant decrease in methylation levels on TP53 3' UTR, providing a more comprehensive evaluation of METTL1's impact on methylation of its targets. The data has been added along with the IGV plots as revised Fig. 1g.

3. Figure 1g: The order of columns should be changed. It is now presented in the opposite order to the column description. The first column should be m⁶A, and the second column should be m⁷G for easy understanding.

Response: We thank the reviewer for the suggestions to facilitate understanding for potential readers. We have switched the columns of m⁶A and m⁷G to match the order mentioned in the results for easy understanding.

4. Pages 112-114: Please describe the properties of all cell lines in more detail. The description, such as "mutant but functional," is not sufficient.

Response: We appreciate the suggestion to provide more information about the TP53 status in the cell lines in our study. We have therefore revised our wording to provide more details for both T98G and LN229 cell lines which harbor mutated TP53. These changes have been implemented at lines 116-120.

5. Extended Data Fig. 1e: It would be better if western blots showing endogenous TP53 protein are included.

Response: We thank the reviewer for the suggestion to include TP53 expression levels. We have added the western blots results to the revised Extended Data Fig. 1e, regarding p53 expression in all the cell lines we used.

6. Figure 2a: Please provide information on the complete list of proteins obtained from mass spectrometry. What are the top-ranked proteins?

Response: We thank the reviewer for the suggestion. We have included a list of proteins enriched by m⁷G probes based on enrichment ratio as the new Supplementary Table S1.

Also, the entire dataset has also been submitted to ProteomeXchange via the PRIDE database. The submission is currently available through reviewer account with username as reviewer_pxd049390@ebi.ac.uk and password as bhDxhEvn. The dataset will be accessible to the public after publication.

7. Extended Data Fig. 2b: What are the K_d values of each protein?

Response: We want to thank the reviewer for the suggestion to quantify the K_d values of each protein binding G vs m⁷G probes. In response, we have calculated and marked the value to the bottom right of each of the gel figures, and we have included the description in the figure legend accordingly.

8. Extended Data Fig. 2d: Provide quantitative data showing the relative binding efficiencies of each variant. What is KH3-4 μ ? Is this the wild-type (full-length) protein lacking KH3-4 or the full-length protein containing point mutations in the KH3-4 domain?

Response: We appreciate the suggestion. As suggested, we have quantified the relative binding efficiency, compared to the wild-type conditions we presented in Fig. 2b, and we have added the results below the original western blotting in Extended Data Fig. 2d. Specifically, KH3-4 refers to the full-length protein with mutations of GxxG to GEEG in the KH domains. We have included detailed information on the constructs in the Results (line 168-169), as well as the figure legend, accordingly.

9. Page 175: “Consistent with the previous report,” cite a proper reference.

Response: We have added the reference accordingly.

10. Page 180: “with most overlapping sites at the 3’ UTR compared to the non-overlapped ones (Fig. 2e, left in blue).” Which line indicates non-overlapped ones in Figure 2e?

Response: The blue line indicates the m⁷G modified sites that do not overlap with IGF2BP protein binding sites. We have included the information in the figure legend to avoid confusion.

11. Extended Data Fig. 4b: Please insert a box plot as an inset, as presented in Figure 3a.

Response: We have added the box plots to Extended Data Fig. 4B as suggested.

12. Extended Data Fig. 4c: For simple comparison, it would be better if the analysis of IGF2BP2 KD data is included in this figure. In addition, how did the authors calculate “IGF2BPs binding intensities” on page 209?

Response: We have included the analysis of IGF2BP2 in Extended Data Fig. 4d accordingly. The binding intensity of IGF2BPs is calculated as the log₂ ratio of enrichment in immunoprecipitated (IP) samples compared to input samples, based on the RNA immunoprecipitation (RIP)-seq data, denoted as log₂(IP/Input). To elaborate, we first measured gene expression levels in both IP and input samples and then used these values to calculate the log₂(IP/Input) ratio, which defines the binding intensity of each IGF2BP protein. We have added more details in the figure legend for reference.

13. Page 210: Please provide IGV plots for the representative transcripts.

Response: We thank the reviewer for pointing out the missing of our IGV plots. We have added the IGV plots for all related targets in HepG2 to revised Extended Data Fig. 4e, all presenting an overlap of m⁷G methylation and IGF2BP3 binding with little signals of m⁶A modification nearby.

14. Figure 3e: Statistical calculations with p-value or r-value are missing.

Response: We want to thank the reviewer for the suggestions on Fig. 3e. We have updated the data without negative signals and include both p-value and PCC analysis results for Fig. 3e, presenting limited overlap of m⁷G and m⁶A modifications.

15. Figure 3: In this figure, the authors characterized and tried to validate different roles of IGF2BPs in m⁶A and m⁷G-modified mRNAs. Although the present data are supportive of the claim, it is still not conclusive. To completely rule out the m⁶A effect on mRNA stability, the reviewer strongly recommends that all (or at least some) experiments should be done under conditions lacking the m⁶A writer. Otherwise, a possible interplay between m⁶A and m⁷G cannot be excluded, although the authors showed an insignificant correlation of methylation levels between m⁷G and m⁶A at the transcript level (Fig. 3e) and the limited overlapping peaks between both modifications in the 3’UTR (Fig. 3f).

Response: We thank the reviewer for this suggestion. To limit the installation of m⁶A on mRNA, we generated a cell line with stable downregulation of METTL3, catalytic component of m⁶A writer complex. With m⁶A methylation inhibited, we further performed the decay assay in

shMETTL3 HepG2 with *IGF2BP1-3* knockdown respectively, to further validate our model where IGF2BP3 regulates mRNA degradation through its interaction with m⁷G modification. We observed a consistent stabilization effect of m⁷G targets upon IGF2BP1 and IGF2BP3 downregulation, while IGF2BP2 still showed limited effect on m⁷G-targets (Fig. 2a). These results all align with the results we observed in wild-type HepG2 cells, suggesting that IGF2BP1 and IGF2BP3 can regulate degradation of m⁷G targets with minimum impact from m⁶A modification. We further confirmed with qPCR analysis that our representative targets present an unaffected promotional effect on mRNA degradation in cells with *METTL3* knockdown compared to control (Fig. 2b). The data has been added to the revised Extended Data Fig. 5f.

Figure 2 for Reviewer 3: m⁶A modification inhibition through stable *METTL3* knockdown showed little impact on the regulation of IGF2BP3 on m⁷G targets. (a) Left: western blotting results confirmed the stable knockdown of *METTL3*; Right, Cumulative of genes half lifetime fold changes (log₂FC) in HepG2 cells with stable knockdown of *METTL3* upon knocking down *IGF2BP1* (left), *IGF2BP2* (middle) and *IGF2BP3* (right). Genes were categorized into two groups according to whether they were marked with m⁷G or not (non-m⁷G). (b) Changes in mRNA levels in HepG2 cells with transient *IGF2BP3* knockdown in cells with stable *METTL3* knockdown compared to control. Mean ± SEM of two independent experiments. Two-tailed Student's t-tests were used. *P* values are marked next to the dots. Calculated half lifetimes are marked in the corresponding colors. (c) Changes in mRNA levels in HepG2 cells upon stable *METTL3* knockdown compared to control with introduction of the dCas13b-IGF2BP3 and guide RNA at the target loci or not (neg). Mean ± SEM of two independent experiments. Two-tailed student's t-tests were used. Calculated half lifetime values are marked in the corresponding colors. *P* values are marked next to the calculated half lifetime values. (d) Changes in mRNA levels in HepG2 cells upon stable *METTL3* knockdown compared to control with introduction of the dCas13b-METTL1 and guide RNA at the target loci or not (neg). Mean ± SEM of two independent experiments. Two-tailed student's t-tests were used. Calculated half lifetime values are marked in the corresponding colors. *P* values are marked next to the calculated half lifetime values.

16. Figure 4: When IGF2BPs are tethered to the mRNAs, one predicts two possible consequences: First, based on IGF2BPs-mediated m⁶A mRNA stabilization, the tethering of IGF2BPs may stabilize the mRNAs. Alternatively, considering that IGF2BPs trigger rapid degradation of m⁷G-modified mRNAs, their tethering may destabilize the mRNAs. Although the authors targeted endogenous mRNAs containing m⁷G residue, some sequences in proximity to the m⁷G residue may be m⁶A-modified. Therefore, as mentioned in comment #15, this reviewer strongly recommends that the tethering experiments should be conducted under conditions depleted of the m⁶A writer.

Response: In frame with the previous comment, we acknowledge the reviewer's suggestion to repeat the tethering assays under the condition with limited m⁶A writer. We therefore introduced dCas13b system to tether either IGF2BP3 or METTL1 to the representative transcripts in sh*METTL3* HepG2 cells and performed decay qPCR to evaluate the impact of m⁶A modification on our model. Interestingly, we observed little impact of METTL3 downregulation as the tethering of IGF2BP3 (Figure 2b) or METTL1 (Figure 2c) consistently promoted degradation of m⁷G-modified targets. We want to thank the reviewer again for the constructive suggestions on experiments, which further supported our conclusion that IGF2BP3 could promote degradation of m⁷G-targets. Both results have been added to the revised Extended Data Fig. 6g-h.

17. Figure 4f: A negative control lane is missing.

Response: We appreciate the suggestion to include negative controls for all the co-IP experiments. We have rerun all the co-IP experiments and included both rabbit IgG as negative control and IGF2BP2, the other IGF2BP family member, as another reference. All the blots (Fig. 4f, Extended Data Fig. 7a and 7b) have been updated accordingly in support of the original conclusions.

18. Figure 4g: The data presented in this figure do not support the authors' claim. Statistical analysis is necessary.

Response: We thank the reviewer for pointing out that necessary statistical analysis was missing. We have added the t-test results for each pair of the last two columns to show that the degradation effects upon IGF2BP3 overexpression can be significantly rescued by *EXOSC2* knockdown, suggesting that IGF2BP3 promoted degradation of its targets through the pathway regulated by *EXOSC2*. We updated both Fig. 4g and Extended Fig. 9f.

19. Figure 5d,e: This reviewer is curious why overexpression or knockdown of IGF2BP3 affects the m7G level of TP53 mRNA? Does this protein directly affect m7G modification?"

Response: We thank the reviewer for pointing out the unclarity of the explanation. Based on the results of our *in vitro* assays, IGF2BP3 can preferentially bind to m⁷G-modified transcripts. Consequently, both sequencing and tethering assays showed that the methylated transcripts are more readily degraded compared to the non-methylated ones. As a result, upon overexpression of IGF2BP3 proteins, the relative methylation level of a certain target of IGF2BP3 decreased as more methylated transcripts were degraded, compared to non-modified ones. To enhance clarity and understanding, we have added explanations in the main text (lines 357-359).

References

1. Huang, H. *et al.* Recognition of RNA N(6)-methyladenosine by IGF2BP proteins enhances mRNA stability and translation. *Nat Cell Biol* **20**, 285-295 (2018).
2. Zhang, L.S. *et al.* Transcriptome-wide Mapping of Internal N(7)-Methylguanosine Methylome in Mammalian mRNA. *Mol Cell* **74**, 1304-1316 e1308 (2019).
3. Wang, X. *et al.* N6-methyladenosine-dependent regulation of messenger RNA stability. *Nature* **505**, 117-120 (2014).
4. Wang, X. *et al.* N(6)-methyladenosine Modulates Messenger RNA Translation Efficiency. *Cell* **161**, 1388-1399 (2015).
5. Zhao, Z. *et al.* QKI shuttles internal m(7)G-modified transcripts into stress granules and modulates mRNA metabolism. *Cell* **186**, 3208-3226 e3227 (2023).
6. Xiao, W. *et al.* Nuclear m(6)A Reader YTHDC1 Regulates mRNA Splicing. *Mol Cell* **61**, 507-519 (2016).
7. Roundtree, I.A. *et al.* YTHDC1 mediates nuclear export of N(6)-methyladenosine methylated mRNAs. *Elife* **6** (2017).
8. Liu, J. *et al.* N (6)-methyladenosine of chromosome-associated regulatory RNA regulates chromatin state and transcription. *Science* **367**, 580-586 (2020).
9. Gao, Z. *et al.* A Comprehensive Analysis of METTL1 to Immunity and Stemness in Pan-Cancer. *Front Immunol* **13**, 795240 (2022).
10. Orellana, E.A. *et al.* METTL1-mediated m(7)G modification of Arg-TCT tRNA drives oncogenic transformation. *Mol Cell* **81**, 3323-3338 e3314 (2021).

REVIEWERS' COMMENTS

Reviewer #1 (Remarks to the Author):

The authors have adequately responded to my inquiries in their response letter. I recommend that the authors include the ART-seq and western blot findings in this manuscript to provide readers with a clearer understanding that IGF2BP3 primarily functions as an m7G reader in regulating RNA decay rather than mRNA translation. Furthermore, it would be helpful if the authors could display the sequences of modified and unmodified RNA probes in Figures 2a (or 2b) and Extended Data Figure 2a.

Reviewer #2 (Remarks to the Author):

This reviewers requests for revision have been satisfied.

Reviewer #3 (Remarks to the Author):

During the revision, Liu et al. successfully addressed all of this reviewer's comments. In particular, the authors clearly demonstrated that IGF2BP3 can promote the degradation of m7G targets regardless of m6A modifications. Therefore, I am confident that the current version is suitable for publication.

Summary

We would like to thank the reviewers again for their time and efforts in reviewing our revised manuscript. We are glad to see that all the reviewers provided quite positive feedback on our last round of revision. The responses are enclosed below.

Reviewer #1 (Remarks to the Author):

The authors have adequately responded to my inquiries in their response letter. I recommend that the authors include the ART-seq and western blot findings in this manuscript to provide readers with a clearer understanding that IGF2BP3 primarily functions as an m⁷G reader in regulating RNA decay rather than mRNA translation. Furthermore, it would be helpful if the authors could display the sequences of modified and unmodified RNA probes in Figures 2a (or 2b) and Extended Data Figure 2a.

Response: We thank the reviewer for the very positive comments and the constructive suggestions.

We have incorporated the ART-seq and western blot findings in the Discussion section (line 437-444) to emphasize the importance of IGF2BP3 in m⁷G-related mRNA decay rather than translation, which further strengthens our conclusion.

We have also included the sequence information of probes in the suggested figures, with detailed description of probe preparation in the Methods section, as suggested by the reviewer.

Reviewer #2 (Remarks to the Author):

This reviewer's requests for revision have been satisfied.

Response: We want to thank the reviewer again for reviewing this manuscript.

Reviewer #3 (Remarks to the Author):

During the revision, Liu et al. successfully addressed all of this reviewer's comments. In particular, the authors clearly demonstrated that IGF2BP3 can promote the degradation of m⁷G targets regardless of m⁶A modifications. Therefore, I am confident that the current version is suitable for publication.

Response: We want to thank the reviewer again for reviewing this manuscript.